# Heavy Ball Momentum for Conditional Gradient

**Bingcong Li**     **Alireza Sadeghi**     **Georgios B. Giannakis**

University of Minnesota - Twin Cities
Minneapolis, MN, USA
{lixx5599, sadeg012, georgios}@umn.edu

## Abstract

Conditional gradient, aka Frank Wolfe (FW) algorithms, have well-documented merits in machine learning and signal processing applications. Unlike projection-based methods, momentum cannot improve the convergence rate of FW, in general. This limitation motivates the present work, which deals with heavy ball momentum, and its impact to FW. Specifically, it is established that heavy ball offers a unifying perspective on the primal-dual (PD) convergence, and enjoys a tighter *per iteration* PD error rate, for multiple choices of step sizes, where PD error can serve as the stopping criterion in practice. In addition, it is asserted that restart, a scheme typically employed jointly with Nesterov's momentum, can further tighten this PD error bound. Numerical results demonstrate the usefulness of heavy ball momentum in FW iterations.

## 1 Introduction

This work studies momentum in Frank Wolfe (FW) methods [9, 10, 16, 20] for solving

$$\min_{\mathbf{x} \in \mathcal{X}} f(\mathbf{x}). \tag{1}$$

Here, $f$ is a convex function with Lipschitz continuous gradients, and the constraint set $\mathcal{X} \subset \mathbb{R}^d$ is assumed convex and compact, where $d$ is the dimension of variable $\mathbf{x}$. Throughout, we let $\mathbf{x}^* \in \mathcal{X}$ denote a minimizer of (1). FW and its variants are prevalent in various machine learning and signal processing applications, such as traffic assignment [12], non-negative matrix factorization [30], video colocation [17], image reconstruction [15], particle filtering [19], electronic vehicle charging [36], recommender systems [11], optimal transport [26], and neural network pruning [34]. The popularity of FW is partially due to the elimination of projection compared with projected gradient descent (GD) [29], leading to computational efficiency especially when $d$ is large. In particular, FW solves a subproblem with a linear loss, i.e., $\mathbf{v}_{k+1} \in \arg\min_{\mathbf{v} \in \mathcal{X}} \langle \nabla f(\mathbf{x}_k), \mathbf{v} \rangle$ at $k$th iteration, and then updates $\mathbf{x}_{k+1}$ as a convex combination of $\mathbf{x}_k$ and $\mathbf{v}_{k+1}$. When dealing with a structured $\mathcal{X}$, a closed-form or efficient solution for $\mathbf{v}_{k+1}$ is available [13, 16], which is preferable over projection.

Unlike projection based algorithms [14, 32] though, momentum does not perform well with FW. Indeed, the lower bound in [16, 20] demonstrates that at least $\mathcal{O}(\frac{1}{\epsilon})$ linear subproblems are required to ensure $f(\mathbf{x}_k) - f(\mathbf{x}^*) \leq \epsilon$, which does not guarantee that momentum is beneficial for FW, because even vanilla FW achieves this lower bound. In this work, we contend that momentum is evidently useful for FW. Specifically, we prove that the *heavy ball momentum* leads to tightened and efficiently computed primal-dual error bound, as well as numerical improvement. To this end, we outline first the primal convergence.

**Primal convergence.** The primal error refers to $f(\mathbf{x}_k) - f(\mathbf{x}^*)$. It is guaranteed for FW that $f(\mathbf{x}_k) - f(\mathbf{x}^*) = \mathcal{O}(1/k), \forall k \geq 1$ [16, 22]. This rate is tight in general since it matches to the lower bound [16, 20]. Other FW variants also ensure the same order of primal error; see e.g., [20, 21].

35th Conference on Neural Information Processing Systems (NeurIPS 2021).

Table 1: A comparison of HFW with relevant works. The "computation" in the third column is short for "the number of required FW subproblems to calculate the PD error per iteration."

| reference | computation | PD conv. type | PD conv. rate |
|:---:|:---:|:---:|:---:|
| [16] | 1 subproblem | Type I | $\frac{27LD^2}{4(K+1)}$ |
| [18] | 2 subproblems | Type II | $\frac{2LD^2}{\sqrt{k+1}}, \forall k$ |
| [28] | 2 subproblems | Type II | $\frac{4LD^2}{k+1}, \forall k$ |
| **This work (Alg. 2)** | 1 subproblem | Type II | $\frac{2LD^2}{k+1}, \forall k$ |
| **This work (Alg. 3)** | 2 subproblems | Type II | $\frac{2LD^2}{k+1+c}, \forall k$ with $c \geq 0$ |

**Primal-dual convergence.** The primal-dual (PD) error quantifies the difference between both the primal and the 'dual' functions from the optimal objective, hence it is an upper bound on the primal error. When the PD error is shown to *converge*, it can be safely used as the stopping criterion: whenever the PD error is less than some prescribed $\epsilon > 0$, $f(\mathbf{x}_k) - f(\mathbf{x}^*) \leq \epsilon$ is ensured automatically. The PD error of FW is convenient to compute, hence FW is suitable for the requirement of "solving problems to some desirable accuracy;" see e.g., [33]. For pruning (two-layer) neural networks [34], the extra training loss incurred by removing neurons can be estimated via the PD error. However, due to technical difficulties, existing analyses on PD error are not satisfactory enough and lack of unification. It is established in [6, 10, 16] that the minimum PD error is sufficiently small, namely $\min_{k \in \{1, \ldots, K\}} \text{PDError}_k = \mathcal{O}\left(\frac{1}{K}\right)$, where $K$ is the total number of iterations. We term such a bound for the minimum PD error as Type I guarantee. Another stronger guarantee, which directly implies Type I bound, emphasizes the per iteration convergence, e.g., $\text{PDError}_k \leq \mathcal{O}\left(\frac{1}{k}\right), \forall k$. We term such guarantees as Type II bound. A Type II bound is reported in [18, Theorem 2], but with an unsatisfactory $k$ dependence. This is improved by [7, 28] with the price of extra computational burden since it involves solving *two* FW subproblems per iteration for computing this PD error. Several related works such as [10] provide a weaker PD error compared with [28]; see a summary in Table 1.

In this work, we show that a computationally affordable Type II bound can be obtained by simply relying on heavy ball momentum. Interestingly, FW based on heavy ball momentum (HFW) also maintains FW's neat geometric interpretation. Through unified analysis, the resultant type II PD error improves over existing bounds; see Table 1. This PD error of HFW is further tightened using *restart*. Although restart is more popular in projection based methods together with Nesterov's momentum [31], we show that restart for FW is natural to adopt jointly with heavy ball. In succinct form, our contributions can be summarized as follows.

• We show through unified analysis that HFW enables a tighter type II guarantee for PD error for multiple choices of the step size. When used as stopping criterion, no extra subproblem is needed.

• The Type II bound can be further tightened by restart triggered through a comparison between two PD-error-related quantities.

• Numerical tests on benchmark datasets support the effectiveness of heavy ball momentum. As a byproduct, a simple yet efficient means of computing local Lipschitz constants becomes available to improve the numerical efficiency of smooth step sizes [13, 22].

**Notation**. Bold lowercase (capital) letters denote column vectors (matrices); $\|\mathbf{x}\|$ stands for a norm of a vector $\mathbf{x}$, whose dual norm is denoted by $\|\mathbf{x}\|_*$; and $\langle \mathbf{x}, \mathbf{y} \rangle$ is the inner product of $\mathbf{x}$ and $\mathbf{y}$.

## 2 Preliminaries

This section outlines FW, starting with standard assumptions that will be taken to hold true throughout.

**Assumption 1.** *(Lipschitz continuous gradient.) The objective function $f : \mathcal{X} \rightarrow \mathbb{R}$ has L-Lipschitz continuous gradients; i.e., $\|\nabla f(\mathbf{x}) - \nabla f(\mathbf{y})\|_* \leq L\|\mathbf{x} - \mathbf{y}\|, \forall \mathbf{x}, \mathbf{y} \in \mathcal{X}$.*

**Assumption 2.** *(Convexity.) The objective function $f : \mathcal{X} \rightarrow \mathbb{R}$ is convex; that is, $f(\mathbf{y}) - f(\mathbf{x}) \geq \langle \nabla f(\mathbf{x}), \mathbf{y} - \mathbf{x} \rangle, \forall \mathbf{x}, \mathbf{y} \in \mathcal{X}$.*

**Assumption 3.** *(Convex and compact constraint set.) The constraint set $\mathcal{X} \subset \mathbb{R}^d$ is convex and compact with diameter $D$, that is, $\|\mathbf{x} - \mathbf{y}\| \leq D, \forall \mathbf{x}, \mathbf{y} \in \mathcal{X}$.*

FW for solving (1) under Assumptions $1 - 3$ is listed in Alg. 1. The subproblem in Line 3 can be visualized geometrically as minimizing a supporting hyperplane of $f(\mathbf{x})$ at $\mathbf{x}_k$, i.e.,

$$\mathbf{v}_{k+1} \in \arg\min_{\mathbf{v} \in \mathcal{X}} f(\mathbf{x}_k) + \langle \nabla f(\mathbf{x}_k), \mathbf{v} - \mathbf{x}_k \rangle.$$

(2)

---

**Algorithm 1** FW [9]

1: **Initialize:** $\mathbf{x}_0 \in \mathcal{X}$
2: **for** $k = 0, 1, \ldots, K - 1$ **do**
3:  $\quad \mathbf{v}_{k+1} = \arg\min_{\mathbf{v} \in \mathcal{X}} \langle \nabla f(\mathbf{x}_k), \mathbf{v} \rangle$
4:  $\quad \mathbf{x}_{k+1} = (1 - \eta_k)\mathbf{x}_k + \eta_k \mathbf{v}_{k+1}$
5: **end for**
6: **Return:** $\mathbf{x}_K$

---

For many constraint sets, efficient implementation or a closed-form solution is available for $\mathbf{v}_{k+1}$; see e.g., [16] for a comprehensive summary. Upon minimizing the supporting hyperplane in (2), $\mathbf{x}_{k+1}$ is updated as a convex combination of $\mathbf{v}_{k+1}$ and $\mathbf{x}_k$ in Line 4 so that no projection is required. The choices on the step size $\eta_k \in [0, 1]$ will be discussed shortly.

The PD error of Alg. 1 is captured by the so-termed *FW gap*, formally defined as

$$\bar{\mathcal{G}}_k := \langle \nabla f(\mathbf{x}_k), \mathbf{x}_k - \mathbf{v}_{k+1} \rangle = \underbrace{f(\mathbf{x}_k) - f(\mathbf{x}^*)}_{\text{primal error}} + \underbrace{f(\mathbf{x}^*) - \min_{\mathbf{v} \in \mathcal{X}} \left[ f(\mathbf{x}_k) + \langle \nabla f(\mathbf{x}_k), \mathbf{v} - \mathbf{x}_k \rangle \right]}_{\text{dual error}} \quad (3)$$

where the second equation is because of (2). It can be verified that both primal and dual errors marked in (3) are no less than 0 by appealing to the convexity of $f$. If $\bar{\mathcal{G}}_k$ converges, one can deduce that the primal error converges. For this reason, $\bar{\mathcal{G}}_k$ is typically used as a stopping criterion for Alg. 1. Next, we focus on the step sizes that ensure convergence.

**Parameter-free step size.** This type of step sizes does not rely on any problem dependent parameters such as $L$ and $D$, and hence it is extremely simple to implement. The most commonly adopted step size is $\eta_k = \frac{2}{k+2}$, which ensures a converging primal error $f(\mathbf{x}_k) - f(\mathbf{x}^*) \leq \frac{2LD^2}{k+1}, \forall k \geq 1$, and a weaker claim on the PD error, $\min_{k \in \{1, \ldots, K\}} \bar{\mathcal{G}}_k = \frac{27LD^2}{4K}$ [16]. A variant of PD convergence has been established recently based on a modified FW gap [28]. Although Type II convergence is observed, the modified FW gap therein is inefficient to serve as stopping criterion because an additional FW subproblem has to be solved per iteration to compute its value.

**Smooth step size.** When the (estimate of) Lipschitz constant $L$ is available, one can adopt the following step sizes in Alg. 1 [22]

$$\eta_k = \min \left\{ \frac{\langle \nabla f(\mathbf{x}_k), \mathbf{x}_k - \mathbf{v}_{k+1} \rangle}{L \|\mathbf{v}_{k+1} - \mathbf{x}_k\|^2}, 1 \right\}. \quad (4)$$

Despite the estimated $L$ is typically too pessimistic to capture the local Lipschitz continuity, such a step size ensures $f(\mathbf{x}_{k+1}) \leq f(\mathbf{x}_k)$; see derivations in Appendix A.1. The PD convergence is studied in [11], where the result is slightly weaker than that of [28].

## 3 FW with heavy ball momentum

After a brief recap of vanilla FW, we focus on the benefits of heavy ball momentum for FW under multiple step size choices, with special emphasis on PD errors.

### 3.1 Prelude

HFW is summarized in Alg. 2. Similar to GD with heavy ball momentum [14, 32], Alg. 2 updates decision variables using a weighted average of gradients $\mathbf{g}_{k+1}$. In addition, the update direction of Alg. 2 is no longer guaranteed to be a descent one. This is because in HFW, $\langle \nabla f(\mathbf{x}_k), \mathbf{x}_k - \mathbf{v}_{k+1} \rangle$ can be negative. Although a stochastic version of heavy ball momentum was adopted in [27] and its variants, e.g., [37], to reduce the mean square error of the gradient estimate, heavy ball is introduced here for a totally different purpose, that is, to improve the PD error. The most significant difference comes at technical perspectives, which is discussed in Sec. 3.4. Next, we gain some intuition on why heavy ball can be beneficial.

Consider $\mathcal{X}$ as an $\ell_2$-norm ball, that is, $\mathcal{X} = \{\mathbf{x} | \|\mathbf{x}\|_2 \leq R\}$. In this case, we have $\mathbf{v}_{k+1} = -\frac{R}{\|\mathbf{g}_{k+1}\|_2} \mathbf{g}_{k+1}$ in Alg. 2. The momentum $\mathbf{g}_{k+1}$ can smooth out the changes of $\{\nabla f(\mathbf{x}_k)\}$, resulting

in a more concentrated sequence $\{\mathbf{v}_{k+1}\}$. Recall that the PD error is closely related to $\mathbf{v}_{k+1}$ [cf. equation (3)]. We hope the "concentration" of $\{\mathbf{v}_{k+1}\}$ to be helpful in reducing the changes of PD error among consecutive iterations so that a Type II PD error bound is attainable.

A few concepts are necessary to obtain a tightened PD error of HFW. First, we introduce the generalized FW gap associated with Alg. 2

---

**Algorithm 2** FW with heavy ball momentum

1: **Initialize:** $\mathbf{x}_0 \in \mathcal{X}, \mathbf{g}_0 = \nabla f(\mathbf{x}_0)$
2: **for** $k = 0, 1, \ldots, K - 1$ **do**
3:     $\mathbf{g}_{k+1} = (1 - \delta_k)\mathbf{g}_k + \delta_k \nabla f(\mathbf{x}_k)$
4:     $\mathbf{v}_{k+1} = \arg\min_{\mathbf{v} \in \mathcal{X}} \langle \mathbf{g}_{k+1}, \mathbf{v} \rangle$
5:     $\mathbf{x}_{k+1} = (1 - \eta_k)\mathbf{x}_k + \eta_k \mathbf{v}_{k+1}$
6: **end for**
7: **Return:** $\mathbf{x}_K$

---

that captures the PD error. Write $\mathbf{g}_{k+1}$ explicitly as $\mathbf{g}_{k+1} = \sum_{\tau=0}^{k} w_k^\tau \nabla f(\mathbf{x}_\tau)$, where $w_k^\tau = \delta_\tau \prod_{j=\tau+1}^{k} (1 - \delta_j) > 0, \ \forall \tau \geq 1$, and $w_k^0 = \prod_{j=1}^{k} (1 - \delta_j) > 0$. Then, define a sequence of linear functions $\{\Phi_k(\mathbf{x})\}$ as

$$\Phi_{k+1}(\mathbf{x}) := \sum_{\tau=0}^{k} w_k^\tau \big[ f(\mathbf{x}_\tau) + \langle \nabla f(\mathbf{x}_\tau), \mathbf{x} - \mathbf{x}_\tau \rangle \big], \ \forall k \geq 0. \tag{5}$$

It is clear that $\Phi_{k+1}(\mathbf{x})$ is a weighted average of the supporting hyperplanes of $f(\mathbf{x})$ at $\{\mathbf{x}_\tau\}_{\tau=0}^{k}$. The properties of $\Phi_{k+1}(\mathbf{x})$, and how they relate to Alg. 2 are summarized in the next lemma.

**Lemma 1.** *For the linear function $\Phi_{k+1}(\mathbf{x})$ in (5), it holds that: i) $\mathbf{v}_{k+1}$ minimizes $\Phi_{k+1}(\mathbf{x})$ over $\mathcal{X}$; and, ii) $f(\mathbf{x}) \geq \Phi_{k+1}(\mathbf{x}), \forall k \geq 0, \forall \mathbf{x} \in \mathcal{X}$.*

From the last lemma, one can see that $\mathbf{v}_k$ is obtained by minimizing $\Phi_k(\mathbf{x})$, which is an affine lower bound on $f(\mathbf{x})$. Hence, HFW admits a geometric interpretation similar to that of FW. In addition, based on $\Phi_k(\mathbf{x})$ we can define the generalized FW gap.

**Definition 1.** *(Generalized FW gap.) The generalized FW gap w.r.t. $\Phi_k(\mathbf{x})$ is*

$$\mathcal{G}_k := f(\mathbf{x}_k) - \min_{\mathbf{x} \in \mathcal{X}} \Phi_k(\mathbf{x}) = f(\mathbf{x}_k) - \Phi_k(\mathbf{v}_k). \tag{6}$$

In words, the generalized FW gap is defined as the difference between $f(\mathbf{x}_k)$ and the minimal value of $\Phi_k(\mathbf{x})$ over $\mathcal{X}$. The newly defined $\mathcal{G}_k$ also illustrates the PD error

$$\mathcal{G}_k = f(\mathbf{x}_k) - \Phi_k(\mathbf{v}_k) = \underbrace{f(\mathbf{x}_k) - f(\mathbf{x}^*)}_{\text{primal error}} + \underbrace{f(\mathbf{x}^*) - \Phi_k(\mathbf{v}_k)}_{\text{dual error}}. \tag{7}$$

For the dual error, we have $f(\mathbf{x}^*) - \Phi_k(\mathbf{v}_k) \geq \Phi_k(\mathbf{x}^*) - \Phi_k(\mathbf{v}_k) \geq 0$, where both inequalities follow from Lemma 1. Hence, $\mathcal{G}_k \geq 0$ automatically serves as an overestimate of both primal and dual errors. When establishing the convergence of $\mathcal{G}_k$, it can be adopted as the stopping criterion for Alg. 2. Related claims have been made for the generalized FW gap [20, 23, 28]. Lack of heavy ball momentum leads to inefficiency, because an additional FW subproblem is needed to compute this gap [28]. Works [20, 23] focus on Nesterov's momentum for FW, that incurs additional memory relative to HFW; see also Sec. 3.4 for additional elaboration. Having defined the generalized FW gap, we next pursue parameter choices that establish Type II convergence guarantees.

### 3.2 Parameter-free step size

We first consider a parameter-free choice for HFW to demonstrate the usefulness of heavy ball

$$\delta_k = \eta_k = \frac{2}{k+2}, \ \forall k \geq 0. \tag{8}$$

Such a choice on $\delta_k$ puts more weight on recent gradients when calculating $\mathbf{g}_{k+1}$, since $w_k^\tau = \mathcal{O}(\frac{\tau}{k^2})$. The following theorem specifies the convergence of $\mathcal{G}_k$.

**Theorem 1.** *If Assumptions 1-3 hold, then choosing $\delta_k$ and $\eta_k$ as in (8), Alg. 2 guarantees that*

$$\mathcal{G}_k = f(\mathbf{x}_k) - \Phi_k(\mathbf{v}_k) \leq \frac{2LD^2}{k+1}, \ \forall k \geq 1.$$

Theorem 1 provides a much stronger PD guarantee for all $k$ than vanilla FW [16, Theorem 2]. In addition to a readily computable generalized FW gap, our rate is tighter than [28], where the provided bound is $\frac{4LD^2}{k+1}$. In fact, the constants in our PD bound even match to the best known primal error of vanilla FW. A direct consequence of Theorem 1 is the convergence of both primal and dual errors.

**Corollary 1.** *Choosing the parameters as in Theorem 1, then $\forall k \geq 1$, Alg.2 guarantees that*

$$\text{primal conv.: } f(\mathbf{x}_k) - f(\mathbf{x}^*) \leq \frac{2LD^2}{k+1}; \quad \text{dual conv.: } f(\mathbf{x}^*) - \Phi_k(\mathbf{v}_k) \leq \frac{2LD^2}{k+1}.$$

*Proof.* Combine Theorem 1 with $f(\mathbf{x}_k) - f(\mathbf{x}^*) \leq \mathcal{G}_k$ and $f(\mathbf{x}^*) - \Phi_k(\mathbf{v}_k) \leq \mathcal{G}_k$ [cf. (7)]. □

### 3.3 Smooth step size

Next, we focus on HFW with a variant of the smooth step size

$$\delta_k = \frac{2}{k+2} \quad \text{and} \quad \eta_k = \max\left\{0, \min\left\{\frac{\langle \nabla f(\mathbf{x}_k), \mathbf{x}_k - \mathbf{v}_{k+1}\rangle}{L\|\mathbf{v}_{k+1} - \mathbf{x}_k\|^2}, 1\right\}\right\}. \tag{9}$$

Comparing with the smooth step size for vanilla FW in (4), it can be deduced that the choice on $\eta_k$ in (9) has to be trimmed to $[0, 1]$ manually. This is because $\langle \nabla f(\mathbf{x}_k), \mathbf{x}_k - \mathbf{v}_{k+1}\rangle$ is no longer guaranteed to be positive. The smooth step size enables an adaptive means of adjusting the weight for $\nabla f(\mathbf{x}_k)$. To see this, note that when $\eta_k = 0$, we have $\mathbf{x}_{k+1} = \mathbf{x}_k$. As a result, $\mathbf{g}_{k+2} = (1 - \delta_{k+1})\mathbf{g}_{k+1} + \delta_{k+1}\nabla f(\mathbf{x}_{k+1}) = (1 - \delta_{k+1})\mathbf{g}_{k+1} + \delta_{k+1}\nabla f(\mathbf{x}_k)$, that is, the weight on $\nabla f(\mathbf{x}_k)$ is adaptively increased to $\delta_k(1 - \delta_{k+1}) + \delta_{k+1}$ if one further unpacks $\mathbf{g}_{k+1}$. Another analytical benefit of the step size in (9) is that it guarantees a non-increasing objective value; see Appendix A.2 for derivations. Convergence of the generalized FW gap is established next.

**Theorem 2.** *If Assumptions 1-3 hold, while $\eta_k$ and $\delta_k$ are chosen as in* (9)*, Alg. 2 guarantees that*

$$\mathcal{G}_k = f(\mathbf{x}_k) - \Phi_k(\mathbf{v}_k) \leq \frac{2LD^2}{k+1}, \; \forall k \geq 1.$$

The proof of Theorem 2 follows from that of Theorem 1 after modifying just one inequality. This considerably simplifies the analysis on the (modified) FW gap compared to vanilla FW with smooth step size [11]. The PD convergence clearly implies the convergence of both primal and dual errors. A similar result to Corollary 1 can be obtained, but we omit it for brevity. We further extend Theorem 2 in Appendix B.4 by showing that if a slightly more difficult subproblem can be solved, it is possible to ensure *per step descent on the PD error; i.e.,* $\mathcal{G}_{k+1} \leq \mathcal{G}_k$.

**Line search.** When choosing $\delta_k = \frac{2}{k+2}$ and $\eta_k$ via line search, HFW can guarantee a Type II PD error of $\frac{2LD^2}{k+1}$; please refer to Appendix B.5 due to space limitation. For completeness, an iterative manner to update $\mathcal{G}_k$ for using as stopping criterion is also described in Appendix C.

### 3.4 Further considerations

There are more choices of $\delta_k$ and $\eta_k$ leading to (primal) convergence. For example, one can choose $\delta_k \equiv \delta \in (0, 1)$ and $\eta_k = \mathcal{O}\left(\frac{1}{k}\right)$ as an extension of [27].[1] A proof is provided in Appendix B.7 for completeness. This analysis framework in [27], however, has two shortcomings: i) the convergence can be only established using $\ell_2$-norm (recall that in Assumption 1, we do not pose any requirement on the norm); and, ii) the final primal error (hence PD error) can only be worse than vanilla FW because their analysis treats $\mathbf{g}_{k+1}$ as $\nabla f(\mathbf{x}_k)$ with errors but not momentum, therefore, it is difficult to obtain the same tight PD bound as in Theorem 1. Our analytical techniques avoid these limitations.

When choosing $\delta_k = \eta_k = \frac{1}{k+1}$, we can recover Algorithm 3 in [1]. Notice that such a choice on $\delta_k$ makes $\mathbf{g}_{k+1}$ a uniform average of all gradients. A slower convergence rate $f(\mathbf{x}_k) - f(\mathbf{x}^*) = \mathcal{O}\left(\frac{LD^2 \ln k}{k}\right)$ was established in [1] through a sophisticated derivation using no-regret online learning. Through our simpler analytical framework, we can attain the same rate while providing more options for the step size.

**Theorem 3.** *Let Assumptions 1-3 hold, and select $\delta_k = \frac{1}{k+1}$ with $\eta_k$ using one of the following options: i) $\eta_k = \frac{1}{k+1}$; ii) as in* (9)*; or iii) line search as in* (26b)*. The generalized FW gap of Alg. 2 then converges with rate*

$$\mathcal{G}_k = f(\mathbf{x}_k) - \Phi_k(\mathbf{v}_k) \leq \frac{LD^2 \ln(k+1)}{2k}, \; \forall k \geq 1.$$

---

[1]We are unable to derive even a primal error bound using the same analysis framework in [27] for step sizes listed in Theorem 1.

The rate in Theorem 3 has worse dependence on $k$ relative to Theorems 1 and 2, partially because too much weight is put on past gradients in $\mathbf{g}_{k+1}$, suggesting that large momentum may not be helpful.

**Heavy ball versus Nesterov's momentum.** A simple rule to compare these two momentums is whether gradient is calculated at the converging sequence $\{\mathbf{x}_k\}$. Heavy ball momentum follows this rule, while Nesterov's momentum computes the gradient at some extrapolation points that are not used in Alg. 2. It is unclear how the original Nesterov's momentum benefits the PD error, but the $\infty$-memory variant of Nesterov's momentum [20, 23, 24], which can be viewed as a combination of heavy ball and Nesterov's momentum, yields a Type II PD error. However, compared with HFW, additional memory should be allocated. In sum, these observations suggest that heavy ball momentum is essentially critical to improve the PD performance of FW. Nesterov's momentum, on the other hand, does not influence PD error when used alone; however, it gives rise to faster (local) primal rates under additional assumptions [20, 23].

### 3.5 A side result: Directional smooth step sizes

Common to both FW and HFW is that the globally estimated $L$ might be too pessimistic for a local update. In this subsection, a local Lipschitz constant is investigated to further improve the numerical efficiency of smooth step sizes in (9). This easily computed local Lipschitz constant is another merit of (H)FW over projection based approaches.

**Definition 2.** *(Directional Lipschitz continuous.) For two points $\mathbf{x}, \mathbf{y} \in \mathcal{X}$, the directional Lipschitz constant $L(\mathbf{x}, \mathbf{y})$ ensures $\|\nabla f(\hat{\mathbf{x}}) - \nabla f(\hat{\mathbf{y}})\|_* \leq L(\mathbf{x}, \mathbf{y})\|\hat{\mathbf{x}} - \hat{\mathbf{y}}\|$ for any $\hat{\mathbf{x}} = (1 - \alpha)\mathbf{x} + \alpha\mathbf{y}, \hat{\mathbf{y}} = (1 - \beta)\mathbf{x} + \beta\mathbf{y}$ with some $\alpha \in [0, 1]$ and $\beta \in [0, 1]$.*

In other words, the directional Lipschitz continuity depicts the local property on the segment between points $\mathbf{x}$ and $\mathbf{y}$. It is clear that $L(\mathbf{x}, \mathbf{y}) \leq L$. Using logistic loss for binary classification as an example, we have $L(\mathbf{x}, \mathbf{y}) \leq \frac{1}{4N} \sum_{i=1}^{N} \frac{\langle \mathbf{a}_i, \mathbf{x} - \mathbf{y} \rangle^2}{\|\mathbf{x} - \mathbf{y}\|_2^2}$, where $N$ is the number of data, and $\mathbf{a}_i$ is the feature of the $i$th datum. As a comparison, the global Lipschitz constant is $L \leq \frac{1}{4N} \sum_{i=1}^{N} \|\mathbf{a}_i\|_2^2$. We show in Appendix E that at least for a class of functions, including widely used logistic loss and quadratic loss, $L(\mathbf{x}, \mathbf{y})$ has an analytical form.

Simply replacing $L$ in (9) with $L(\mathbf{x}_k, \mathbf{v}_{k+1})$, i.e.,

$$\eta_k = \max\left\{0, \min\left\{\frac{\langle \nabla f(\mathbf{x}_k), \mathbf{x}_k - \mathbf{v}_{k+1}\rangle}{L(\mathbf{x}_k, \mathbf{v}_{k+1})\|\mathbf{v}_{k+1} - \mathbf{x}_k\|^2}, 1\right\}\right\} \tag{10}$$

we can obtain what we term *directionally smooth step size*. Upon exploring the collinearity of $\mathbf{x}_k$, $\mathbf{x}_{k+1}$ and $\mathbf{v}_{k+1}$, a simple modification of Theorem 2 ensures the PD convergence.

**Corollary 2.** *Choosing $\delta_k = \frac{2}{k+2}$, and $\eta_k$ via (10), Alg. 2 ensures*

$$\mathcal{G}_k = f(\mathbf{x}_k) - \Phi_k(\mathbf{v}_k) \leq \frac{2LD^2}{k+1}, \ \forall k \geq 1.$$

The directional Lipschitz constant can also replace the global one in other FW variants, such as [13, 22], with theories therein still holding. As we shall see in numerical tests, directional smooth step sizes outperform the vanilla one by an order of magnitude.

## 4 Restart further tightens the PD error

Up till now it is established that the heavy ball momentum enables a unified analysis for tighter Type II PD bounds. In this section, we show that if the computational resources are sufficient for solving two FW subproblems per iteration, the PD error can be further improved by restart when the standard FW gap is smaller than generalized FW gap. Restart is typically employed by Nesterov's momentum in projection based methods [31] to cope with the robustness to parameter estimates, and to capture the local geometry of problem (1). However, it is natural to integrate restart with heavy ball momentum in FW regime. In addition, restart provides an answer to the following question: *which is smaller, the generalized FW gap or the vanilla one?* Previous works using the generalized FW gap have not addressed this question [20, 23, 28].

---

**Algorithm 3** FW with heavy ball momentum and restart

---
1: **Initialize:** $\mathbf{x}_0^0 \in \mathcal{X}, \mathbf{g}_0^0 = \nabla f(\mathbf{x}_0^0), s \leftarrow 0, C^0 = 0, \mathcal{G}_0^0 = \bar{\mathcal{G}}_0^0$
2: **while** [not terminated] **do**
3: $\quad$ $k \leftarrow 0, \mathbf{g}_0^s = \nabla f(\mathbf{x}_0^s)$
4: $\quad$ **while** [$\mathcal{G}_k^s \leq \bar{\mathcal{G}}_k^s$ or $k = 0$] and [not terminated] **do** $\qquad$ ▷ Check whether restart is needed
5: $\quad\quad$ $\delta_k^s = \frac{2}{k+2+C^s}$
6: $\quad\quad$ $\mathbf{g}_{k+1}^s = (1 - \delta_k^s)\mathbf{g}_k^s + \delta_k^s \nabla f(\mathbf{x}_k^s)$
7: $\quad\quad$ $\mathbf{v}_{k+1}^s = \arg\min_{\mathbf{x} \in \mathcal{X}} \langle \mathbf{g}_{k+1}^s, \mathbf{x} \rangle$
8: $\quad\quad$ $\mathbf{x}_{k+1}^s = (1 - \eta_k^s)\mathbf{x}_k^s + \eta_k^s \mathbf{v}_{k+1}^s$
9: $\quad\quad$ $\bar{\mathbf{v}}_{k+1}^s = \arg\min_{\mathbf{x} \in \mathcal{X}} \langle \nabla f(\mathbf{x}_{k+1}^s), \mathbf{x} \rangle$
10: $\quad\quad$ $\mathcal{G}_{k+1}^s = f(\mathbf{x}_{k+1}^s) - \Phi_{k+1}^s(\mathbf{v}_{k+1}^s)$ $\qquad\qquad\qquad$ ▷ Generalized FW gap
11: $\quad\quad$ $\bar{\mathcal{G}}_{k+1}^s = \langle \nabla f(\mathbf{x}_k^s), \mathbf{x}_k^s - \bar{\mathbf{v}}_{k+1}^s \rangle$ $\qquad\qquad\qquad$ ▷ Vanilla FW gap
12: $\quad\quad$ $k \leftarrow k + 1$
13: $\quad$ **end while**
14: $\quad$ $K_s \leftarrow k, \mathbf{x}_0^{s+1} = \mathbf{x}_{K_s}^s, C^{s+1} = \frac{2LD^2}{\mathcal{G}_{K_s}^s}, s \leftarrow s + 1$
15: **end while**

---

FW with heavy ball momentum and restart is summarized under Alg. 3. For exposition clarity, when updating the counters such as $k$ and $s$, we use notation '$\leftarrow$'. Alg. 3 contains two loops. The inner loop is the same as Alg. 2 except for computing a standard FW gap (Line 11) in addition to the generalized one (Line 10). The variable $K_s$, depicting the iteration number of inner loop $s$, is of analysis purpose. Alg. 3 can be terminated immediately whenever $\min\{\mathcal{G}_k^s, \bar{\mathcal{G}}_k^s\} \leq \epsilon$ for a desirable $\epsilon > 0$. The restart happens when the standard FW gap is smaller than generalized FW gap. And after restart, $\mathbf{g}_{k+1}^s$ will be reset. For Alg. 3, the linear functions used for generalized FW gap are defined stage-wisely

$$\Phi_0^s(\mathbf{x}) = f(\mathbf{x}_0^s) + \langle \nabla f(\mathbf{x}_0^s), \mathbf{x} - \mathbf{x}_0^s \rangle \tag{11a}$$

$$\Phi_{k+1}^s(\mathbf{x}) = (1 - \delta_k^s)\Phi_k^s(\mathbf{x}) + \delta_k^s [f(\mathbf{x}_k^s) + \langle \nabla f(\mathbf{x}_k^s), \mathbf{x} - \mathbf{x}_k^s \rangle], \forall k \geq 0. \tag{11b}$$

It can be verified that $\mathbf{v}_{k+1}^s$ minimizes $\Phi_{k+1}^s(\mathbf{x})$ over $\mathcal{X}$ for any $k \geq 0$. In addition, we also have $f(\mathbf{x}_0^s) - \Phi_0^s(\mathbf{v}_0^s) = \bar{\mathcal{G}}_{K_{s-1}}^{s-1}$ where $\mathbf{v}_0^s = \arg\min_{\mathbf{x} \in \mathcal{X}} \Phi_0^s(\mathbf{x})$.

There are two tunable parameters $\eta_k^s$ and $\delta_k^s$. The choice on $\delta_k^s$ has been provided directly in Line 5, where it is adaptively decided using a variable $C^s$ relating to the generalized FW gap. Three choices are readily available for $\eta_k^s$: i) $\eta_k^s = \delta_k^s$; ii) smooth step size:

$$\eta_k^s = \max\left\{0, \min\left\{\frac{\langle \nabla f(\mathbf{x}_k^s), \mathbf{x}_k^s - \mathbf{v}_{k+1}^s \rangle}{L\|\mathbf{v}_{k+1}^s - \mathbf{x}_k^s\|^2}, 1\right\}\right\}; \tag{12}$$

and, iii) line search

$$\eta_k^s = \arg\min_{\eta \in [0,1]} f\big((1 - \eta)\mathbf{x}_k^s + \eta \mathbf{v}_{k+1}^s\big). \tag{13}$$

Note that the directionally smooth step size, i.e., replacing $L$ with $L(\mathbf{x}_k^s, \mathbf{v}_{k+1}^s)$ in (12) is also valid for convergence. We omit it to reduce repetition. Next we show how restart improves the PD error.

**Theorem 4.** *Choose $\eta_k^s$ via one of the three manners: i) $\eta_k^s = \delta_k^s$; ii) as in (12); or iii) as in (13). If there is no restart (e.g., $s = 0$ when terminating), then Alg. 3 guarantees that*

$$\mathcal{G}_k^0 = f(\mathbf{x}_k^0) - \Phi_k(\mathbf{v}_k^0) \leq \frac{2LD^2}{k+1}, \forall k \geq 1. \tag{14a}$$

*If restart happens, in additional to (14a), we have*

$$\mathcal{G}_k^s = f(\mathbf{x}_k^s) - \Phi_k(\mathbf{v}_k^s) < \frac{2LD^2}{k+C^s}, \forall k \geq 1, \forall s \geq 1, \; with \; C^s \geq 1 + \sum_{j=0}^{s-1} K_j. \tag{14b}$$

Besides the convergence of both primal and dual errors of Alg. 3, Theorem 4 implies that when no restart happens, the generalized FW gap is smaller than the standard one, demonstrating that the

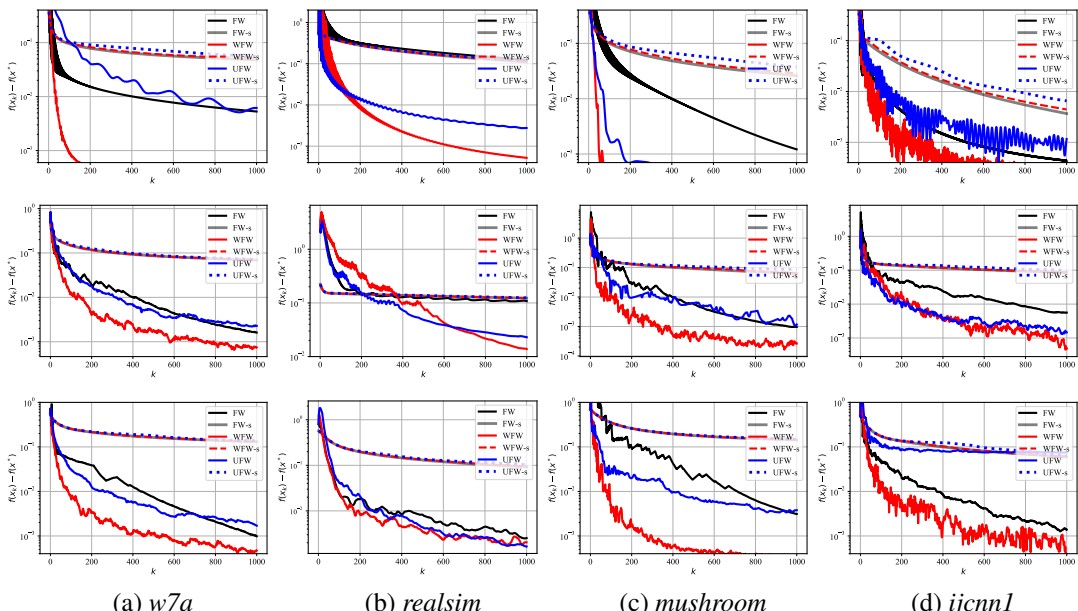

|  |  |  |  |
|---|---|---|---|
| (a) *w7a* | (b) *realsim* | (c) *mushroom* | (d) *ijcnn1* |

Figure 1: Performance of FW variants for binary classification with the constraint being an $\ell_2$-norm ball (first row), an $\ell_1$-norm ball (second row), and an $n$-support norm ball (third row).

former is more suitable for the purpose of "stopping criterion". When restarted, Theorem 4 provides a strictly improved bound compared with Theorems 1, 2, and 6, since the denominator of the RHS in (14b) is no smaller than the total iteration number. An additional comparison with [28], where two subproblems are also required, once again confirms the power of heavy ball momentum to improve the constants in the PD error rate, especially with the aid of restart. The restart scheme (with slight modification) can also be employed in [23, 24, 28] to tighten their PD error.

## 5 Numerical tests

This section presents numerical tests to showcase the effectiveness of HFW on different machine learning problems. Since there are two parameters' choices for HFW in Theorems 1 and 3, we term them as weighted FW (WFW) and uniform FW (UFW), respectively, depending on the weight of $\{\nabla f(\mathbf{x}_k)\}$ in $\mathbf{g}_{k+1}$. When using smooth step size, the corresponding algorithms are marked as WFW-s and UFW-s. For comparison, the benchmark algorithms include: FW with $\eta_k = \frac{2}{k+2}$ (FW); and, FW with smooth step size (FW-s) in (4).

### 5.1 Binary classification

We first test the performance of Alg. 2 on binary classification using logistic regression

$$f(\mathbf{x}) = \frac{1}{N} \sum_{i=1}^{N} \ln \left(1 + \exp(-b_i \langle \mathbf{a}_i, \mathbf{x} \rangle)\right). \tag{15}$$

Here $(\mathbf{a}_i, b_i)$ is the (feature, label) pair of datum $i$, and $N$ is the number of data. Datasets from LIBSVM[2] are used in the numerical tests, where details of the datasets are deferred to Appendix F due to space limitation.

$\ell_2$-**norm ball constraint.** We start with $\mathcal{X} = \{\mathbf{x} | \|\mathbf{x}\|_2 \leq R\}$. The primal errors are plotted in the first row of Fig. 1. We use primal error here for a fair comparison. It can be seen that the parameter-free step sizes achieve better performance compared with the smooth step sizes mainly because the quality of $L$ estimate. Such a problem can be relived through directional smooth step sizes as we shall shortly. Among parameter-free step sizes, it can be seen that WFW consistently

---

[2]https://www.csie.ntu.edu.tw/~cjlin/libsvmtools/datasets/binary.html.

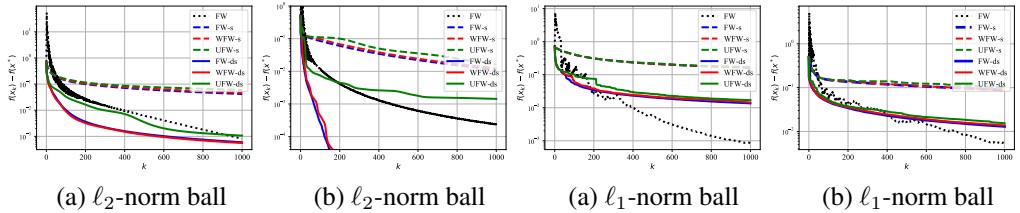

| (a) $\ell_2$-norm ball | (b) $\ell_2$-norm ball | (a) $\ell_1$-norm ball | (b) $\ell_1$-norm ball |

Figure 2: Performance of directionally smooth step sizes. (a) and (c) are tested on *mushroom*; and (b) and (d) use *ijcnn1*.

outperforms both UFW and FW on all tested datasets, while UFW converges faster than FW only on datasets *realsim* and *mushroom*. For smooth step sizes, the per-step-descent property is validated. The excellent performance of HFW can be partially explained by the similarity of its update, namely $\mathbf{x}_{k+1} = (1 - \eta_k)\mathbf{x}_k + \eta_k R \frac{\mathbf{g}_{k+1}}{\|\mathbf{g}_{k+1}\|_2}$, with normalized gradient descent (NGD) one, that is given by $\mathbf{x}_{k+1} = \mathrm{Proj}_{\mathcal{X}} \left( \mathbf{x}_k - \eta_k \frac{\mathbf{g}_{k+1}}{\|\mathbf{g}_{k+1}\|_2} \right)$. However, there is also a subtle difference between HFW and NGD updates. Indeed, when projection is in effect, $\mathbf{x}_{k+1}$ in NGD will lie on the boundary of the $\ell_2$-norm ball. Due to the convex combination nature of the update in HFW, it is unlikely to have $\mathbf{x}_{k+1}$ on the boundary, though it can come arbitrarily close.

**$\ell_1$-norm ball constraint.** Here $\mathcal{X} = \{\mathbf{x} | \|\mathbf{x}\|_1 \leq R\}$ denotes the constraint set that promotes sparse solutions. In the simulation, $R$ is tuned for a solution with similar sparsity as the dataset itself. The results are showcased in the second row of Fig. 1. For smooth step sizes, FW-s, UFW-s, and WFW-s exhibit similar performances, and their curves are smooth. On the other hand, parameter-free step sizes eventually outperform smooth step sizes though the curves zig-zag. (The curves on *realsim* are smoothed to improve figure quality.) UFW has similar performance on *w7a* and *mushroom* with FW and faster convergence on other datasets. Once again, WFW consistently outperforms FW and UFW.

**$n$-support norm ball constraint.** The $n$-support norm ball is a tighter relaxation of a sparsity enforcing $\ell_0$-norm ball combined with an $\ell_2$-norm penalty compared with ElasticNet [38]. It gives rise to $\mathcal{X} = \mathrm{conv}\{\mathbf{x} | \|\mathbf{x}\|_0 \leq n, \|\mathbf{x}\|_2 \leq R\}$, where $\mathrm{conv}\{\cdot\}$ denotes the convex hull [3]. The closed-form solution of $\mathbf{v}_{k+1}$ is given in [25]. In the simulation, we choose $n = 2$ and tune $R$ for a solution whose sparsity is similar to the adopted dataset. The results are showcased in the third row of Fig. 1. For smooth step sizes, FW-s and WFW-s exhibit similar performance, while UFW-s converges slightly slower on *ijcnn1*. Regarding parameter-free step sizes, UFW does not offer faster convergence compared with FW on the tested datasets, but WFW again has numerical merits.

**Directionally smooth step sizes.** The results in Fig. 2 validate the effectiveness of directionally smooth (-ds) step sizes. For all datasets tested, the benefit of adopting $L(\mathbf{x}_k, \mathbf{v}_{k+1})$ is evident, as it improves the performance of smooth step sizes by an order of magnitude. In addition, it is also observed that UFW-ds performs worse than WFW-ds, which suggests that putting too much weight on past gradients could be less attractive in practice.

**Additional comparisons.** We also compare HFW with a generalized version of [27], where we set $\delta_k = \delta \in (0, 1), \forall k$ in Alg. 2. Two specific choices, i.e., $\delta = 0.6$, and $\delta = 0.8$, are plotted in Fig. 3, where the $\ell_2$-norm ball and $n$-support norm ball are adopted as constraints. In both cases, WFW converges faster than the algorithm adapted from [27]. In addition, the choice of $\delta$ has major impact on convergence behavior, while WFW avoids this need for manual tuning of $\delta$. The performance of WFW with restart, i.e., Alg. 3, is also shown in Fig. 3. Although it slightly outperforms WFW,

restart also doubles the computational burden due to the need of solving two FW subproblems. From this point of view, WFW with restart is more of theoretical rather than practical interest. In addition, it is observed that Alg. 3 is not restarted after the first few iterations, which suggests that the generalized FW gap is smaller than the vanilla one, at least in the early stage of convergence. Thus, the generalized FW gap is attractive as a stopping criterion when a solution with moderate accuracy is desirable.

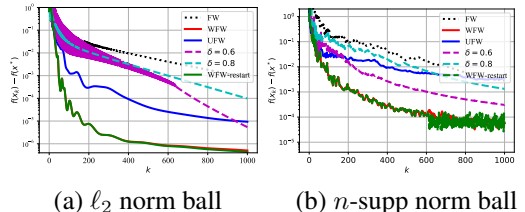

| (a) $\ell_2$ norm ball | (b) $n$-supp norm ball |

Figure 3: Comparison of HFW with other algorithms on *muchroom*.

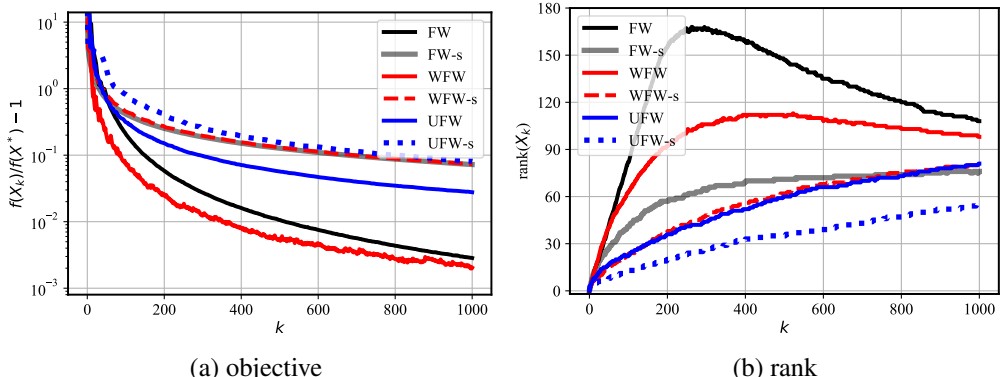

|(a) objective|(b) rank|

Figure 4: Performance of FW variants for matrix completion on *MovieLens100K*.

In a nutshell, the numerical experiments suggest that heavy ball momentum performs best with parameter-free step sizes with the momentum weight carefully adjusted. WFW is mainly recommended because it achieves improved empirical performance compared to UFW and FW, regardless of the constraint sets. The smooth step sizes on the other hand, eliminate the zig-zag behavior at the price of convergence slowdown due to the need of $L$, while directionally smooth step sizes can be helpful to alleviate this convergence slowdown.

## 5.2 Matrix completion

This subsection focuses on matrix completion problems for recommender systems. Consider a matrix $\mathbf{A} \in \mathbb{R}^{m \times n}$ with partially observed entries, i.e., entries $A_{ij}$ for $(i,j) \in \mathcal{K}$ are known, where $\mathcal{K} \subset \{1, \ldots, m\} \times \{1, \ldots, n\}$. Based on the observed entries that can be contaminated by noise, the goal is to predict the missing entries. Within the scope of recommender systems, a commonly adopted empirical observation is that $\mathbf{A}$ is low rank [4, 5, 8], leading to the following problem formulation.

$$\min_{\mathbf{X}} \quad \frac{1}{2} \sum_{(i,j) \in \mathcal{K}} (X_{ij} - A_{ij})^2 \quad \text{s.t. } \|\mathbf{X}\|_{\text{nuc}} \leq R. \tag{16}$$

Problem (16) is difficult to solve using GD because projection onto a nuclear norm ball requires a full SVD, which has complexity $\mathcal{O}\big(mn(m \wedge n)\big)$ with $(m \wedge n) := \min\{m, n\}$. In contrast, FW and its variants are more suitable for (16) since the FW subproblem has complexity less than $\mathcal{O}(mn)$ [2].

Heavy ball based FW are tested using dataset *MovieLens100K*[3]. Following the initialization of [11], the numerical results can be found in Fig. 4. Subfigures (a) and (b) depict the optimality error and rank versus $k$ for $R = 3$. For parameter-free step sizes, WFW converges faster than FW while finding solutions with lower rank. The low rank solution of UFW is partially because it does not converge sufficiently. For smooth step sizes, UFW-s finds a solution with slightly larger objective value but much lower rank compared with WFW-s and FW-s. Overall, when a small optimality error is the priority, WFW is more attractive; while UFW-s is useful for finding low rank solutions.

## 6 Conclusions and future directions

This work demonstrated the merits of heavy ball momentum for FW. Multiple choices of the step size ensured a tighter Type II primal-dual error bound that can be efficiently computed when adopted as stopping criterion. An even tighter PD error bound can be achieved by relying jointly on heavy ball momentum and restart. A novel and general approach was developed to compute local Lipschitz constants in FW type algorithms. Numerical tests in the paradigms of logistic regression and matrix completion demonstrated the effectiveness of heavy ball momentum in FW.

Our future research agenda includes performance evaluation of heavy ball momentum for various learning tasks. For example, HFW holds great potential when fairness is to be accounted for [35].

---

[3]`https://grouplens.org/datasets/movielens/100k/`

**Acknowledgement**

This work is supported by NSF grants 1901134, 2126052, and 2128593. The authors would also like to thank the anonymous reviewers for their feedback.

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
