# Supplementary Document for
# "Heavy Ball Momentum for Conditional Gradient"

## A    Preludes

### A.1    $f(\mathbf{x}_{k+1}) \leq f(\mathbf{x}_k)$ for the smooth step sizes in Alg. 1

When using the step size (4) in Alg. 1, $f(\mathbf{x}_{k+1}) \leq f(\mathbf{x}_k)$ is ensured automatically. To see this, we have from Assumption 1 that

$$f(\mathbf{x}_{k+1}) - f(\mathbf{x}_k) \leq \langle \nabla f(\mathbf{x}_k), \mathbf{x}_{k+1} - \mathbf{x}_k \rangle + \frac{L}{2}\|\mathbf{x}_{k+1} - \mathbf{x}_k\|^2 \tag{17}$$

$$\overset{(a)}{=} \eta_k \langle \nabla f(\mathbf{x}_k), \mathbf{v}_{k+1} - \mathbf{x}_k \rangle + \frac{\eta_k^2 L}{2}\|\mathbf{v}_{k+1} - \mathbf{x}_k\|^2 \overset{(b)}{\leq} 0$$

where (a) uses $\mathbf{x}_{k+1} = (1 - \eta_k)\mathbf{x}_k + \eta_k \mathbf{v}_{k+1}$; and (b) is because $\eta_k$ minimizes the RHS of (17) over $[0, 1]$.

### A.2    $f(\mathbf{x}_{k+1}) \leq f(\mathbf{x}_k)$ for the smooth step sizes in Alg. 2

When using the step size (10) in Alg. 2, $f(\mathbf{x}_{k+1}) \leq f(\mathbf{x}_k)$ is ensured.

$$f(\mathbf{x}_{k+1}) - f(\mathbf{x}_k) \leq \eta_k \langle \nabla f(\mathbf{x}_k), \mathbf{v}_{k+1} - \mathbf{x}_k \rangle + \frac{\eta_k^2 L}{2}\|\mathbf{v}_{k+1} - \mathbf{x}_k\|^2 \leq 0$$

where the last ineqaulity is because $\eta_k$ minimizes $\eta \langle \nabla f(\mathbf{x}_k), \mathbf{v}_{k+1} - \mathbf{x}_k \rangle + \frac{\eta^2 L}{2}\|\mathbf{v}_{k+1} - \mathbf{x}_k\|^2$ over $[0, 1]$.

## B    Missing proofs in Section 3.

### B.1    Proof of Lemma 1

*Proof.* Using $\mathbf{g}_{k+1} = \sum_{\tau=0}^{k} w_k^\tau \nabla f(\mathbf{x}_\tau)$, we have

$$\arg\min_{\mathbf{x} \in \mathcal{X}} \Phi_{k+1}(\mathbf{x}) = \arg\min_{\mathbf{x} \in \mathcal{X}} \Big\langle \sum_{\tau=0}^{k} w_k^\tau \nabla f(\mathbf{x}_\tau), \mathbf{x} \Big\rangle = \arg\min_{\mathbf{x} \in \mathcal{X}} \langle \mathbf{g}_{k+1}, \mathbf{x} \rangle.$$

By comparing with Line 4 of Alg. 2, one can see that $\mathbf{v}_{k+1}$ is a minimizer of $\Phi_{k+1}(\mathbf{x})$ over $\mathcal{X}$. To prove that $\Phi_{k+1}(\mathbf{x})$ is a lower bound of $f(\mathbf{x})$, we appeal to convexity to write

$$\Phi_{k+1}(\mathbf{x}) = \sum_{\tau=0}^{k} w_k^\tau \big[ f(\mathbf{x}_\tau) + \langle \nabla f(\mathbf{x}_\tau), \mathbf{x} - \mathbf{x}_\tau \rangle \big] \leq \sum_{\tau=0}^{k} w_k^\tau f(\mathbf{x}) = f(\mathbf{x})$$

where the last equation is because $\sum_{\tau=0}^{k} w_k^\tau = 1$ holds for any $k$. The proof is thus complete.    □

### B.2    Proof of Theorem 1

*Proof.* Using Assumption 1, we have

$$f(\mathbf{x}_{k+1}) - f(\mathbf{x}_k) \tag{18}$$

$$\leq \langle \nabla f(\mathbf{x}_k), \mathbf{x}_{k+1} - \mathbf{x}_k \rangle + \frac{L}{2}\|\mathbf{x}_{k+1} - \mathbf{x}_k\|^2$$

$$= \eta_k \langle \nabla f(\mathbf{x}_k), \mathbf{v}_{k+1} - \mathbf{x}_k \rangle + \frac{\eta_k^2 L}{2}\|\mathbf{v}_{k+1} - \mathbf{x}_k\|^2.$$

Inequality (18) is standard in the analysis of FW and its variants. Letting $\Phi_0(\mathbf{x}) \equiv 0$, and $\mathbf{v}_0$ be any point in $\mathcal{X}$, it can be verified that $\Phi_{k+1}(\mathbf{x}) = (1 - \delta_k)\Phi_k(\mathbf{x}) + \delta_k \big[ f(\mathbf{x}_k) + \langle \nabla f(\mathbf{x}_k), \mathbf{x} - \mathbf{x}_k \rangle \big]$,

from which we have

$$\Phi_{k+1}(\mathbf{v}_{k+1}) \tag{19}$$

$$= (1 - \delta_k)\Phi_k(\mathbf{v}_{k+1}) + \delta_k\Big[f(\mathbf{x}_k) + \big\langle\nabla f(\mathbf{x}_k), \mathbf{v}_{k+1} - \mathbf{x}_k\big\rangle\Big]$$

$$\overset{(a)}{\geq} (1 - \delta_k)\Phi_k(\mathbf{v}_k) + \delta_k\Big[f(\mathbf{x}_k) + \big\langle\nabla f(\mathbf{x}_k), \mathbf{v}_{k+1} - \mathbf{x}_k\big\rangle\Big]$$

where (a) is because $1 - \delta_k \geq 0$ and $\mathbf{v}_k$ minimizes $\Phi_k(\mathbf{x})$ over $\mathcal{X}$ (hence $\Phi_k(\mathbf{v}_k) \leq \Phi_k(\mathbf{v}_{k+1})$).
Now subtracting $\Phi_{k+1}(\mathbf{v}_{k+1})$ on both sides of (18), we have

$$f(\mathbf{x}_{k+1}) - \Phi_{k+1}(\mathbf{v}_{k+1}) \tag{20}$$

$$\overset{(b)}{\leq} (1 - \delta_k)\big[f(\mathbf{x}_k) - \Phi_k(\mathbf{v}_k)\big] + \frac{\delta_k^2 L\|\mathbf{v}_{k+1} - \mathbf{x}_k\|^2}{2}$$

$$\overset{(c)}{\leq} (1 - \delta_k)\big[f(\mathbf{x}_k) - \Phi_k(\mathbf{v}_k)\big] + \frac{\delta_k^2 LD^2}{2}$$

where (b) uses $\eta_k = \delta_k$ and (19); and (c) relies on Assumption 3. For convenience, let $\Delta(i,j) := \prod_{\tau=i}^{j}(1 - \delta_\tau)$, and unroll (20) to arrive at

$$f(\mathbf{x}_{k+1}) - \Phi_{k+1}(\mathbf{v}_{k+1})$$

$$\leq \Delta(0,k)\big[f(\mathbf{x}_0) - \Phi_0(\mathbf{v}_0)\big] + \sum_{\tau=0}^{k}\frac{LD^2\delta_\tau^2}{2}\Delta(\tau+1, k).$$

Plugging in the values of $\delta_k$ completes the proof. $\qquad\square$

## B.3 Proof of Theorem 2

*Proof.* The first a few steps are the same as the proof of Theorem 1; i.e., we have (18) and (19).
Combining (18) and (19), we arrive at

$$f(\mathbf{x}_{k+1}) - \Phi_{k+1}(\mathbf{v}_{k+1}) \tag{21}$$

$$\leq (1 - \delta_k)\big[f(\mathbf{x}_k) - \Phi_k(\mathbf{v}_k)\big] + (\eta_k - \delta_k)\big\langle\nabla f(\mathbf{x}_k), \mathbf{v}_{k+1} - \mathbf{x}_k\big\rangle + \frac{\eta_k^2 L\|\mathbf{v}_{k+1} - \mathbf{x}_k\|^2}{2}.$$

It can be verified that the specific choice of $\eta_k$ minimizes the RHS of (21) over $[0, 1]$. Hence we have

$$f(\mathbf{x}_{k+1}) - \Phi_{k+1}(\mathbf{v}_{k+1}) \tag{22}$$

$$\leq (1 - \delta_k)\big[f(\mathbf{x}_k) - \Phi_k(\mathbf{v}_k)\big] + \frac{\eta_k^2 L\|\mathbf{v}_{k+1} - \mathbf{x}_k\|^2}{2} + (\eta_k - \delta_k)\big\langle\nabla f(\mathbf{x}_k), \mathbf{v}_{k+1} - \mathbf{x}_k\big\rangle$$

$$\overset{(a)}{\leq} (1 - \delta_k)\big[f(\mathbf{x}_k) - \Phi_k(\mathbf{v}_k)\big] + \frac{\alpha_k^2 L\|\mathbf{v}_{k+1} - \mathbf{x}_k\|^2}{2} + (\alpha_k - \delta_k)\big\langle\nabla f(\mathbf{x}_k), \mathbf{v}_{k+1} - \mathbf{x}_k\big\rangle$$

$$\overset{(b)}{=} (1 - \delta_k)\big[f(\mathbf{x}_k) - \Phi_k(\mathbf{v}_k)\big] + \frac{\delta_k^2 L\|\mathbf{v}_{k+1} - \mathbf{x}_k\|^2}{2}$$

$$\leq \big[f(\mathbf{x}_0) - \Phi_0(\mathbf{v}_0)\big]\prod_{\tau=0}^{k}(1 - \delta_\tau) + \sum_{\tau=0}^{k}\frac{LD^2\delta_\tau^2}{2}\prod_{j=\tau+1}^{k}(1 - \delta_j)$$

$$\leq \frac{2LD^2}{k+2}$$

where in (a) $\alpha_k$ can be chosen as any number in $[0, 1]$; in (b) we set $\alpha_k = \delta_k$. This completes the
proof. $\qquad\square$

## B.4 An extension of Theorem 2 for per step descent of $\mathcal{G}_k$

In this section, we show that it is possible to ensure per step descent on generalized FW gap when a
more difficult subproblem can be solved. In particular, we will replace Line 4 of Alg. 2 and choose

parameters as

$$(\delta_k, \mathbf{v}_{k+1}) = \underset{\delta \in [0,1], \mathbf{v} \in \mathcal{X}}{\arg\min} (1 - \delta)\big[f(\mathbf{x}_k) - \Phi_k(\mathbf{v}_k)\big] + \frac{\delta^2 L \|\mathbf{v} - \mathbf{x}_k\|^2}{2} \tag{23a}$$

$$\eta_k = \delta_k. \tag{23b}$$

It is clear that (23a) is harder to solve compared with a FW subproblem. The choice of $\delta_k$ enables an adaptive weights for $\nabla f(\mathbf{x}_k)$ in $\mathbf{g}_{k+1}$. Next we present the main result for such a parameter choice.

**Theorem 5.** *When Assumptions 1, 2 and 3 are satisfied, choosing* $\mathbf{v}_{k+1}$, $\eta_k$ *and* $\delta_k$ *according to* (23), *Alg. 2 guarantees that: i)* $\mathcal{G}_{k+1} \le \mathcal{G}_k$, *and ii)*

$$\mathcal{G}_k = f(\mathbf{x}_k) - \Phi_k(\mathbf{v}_k) \le \frac{2LD^2}{k+1}, \ \forall k \ge 1.$$

*Proof.* It can be seen that (21) still holds, from which we have

$$f(\mathbf{x}_{k+1}) - \Phi_{k+1}(\mathbf{v}_{k+1}) \tag{24}$$

$$\le (1 - \delta_k)\big[f(\mathbf{x}_k) - \Phi_k(\mathbf{v}_k)\big] + \frac{\eta_k^2 L \|\mathbf{v}_{k+1} - \mathbf{x}_k\|^2}{2} + (\eta_k - \delta_k)\langle \nabla f(\mathbf{x}_k), \mathbf{v}_{k+1} - \mathbf{x}_k \rangle$$

$$\overset{(a)}{=} (1 - \delta_k)\big[f(\mathbf{x}_k) - \Phi_k(\mathbf{v}_k)\big] + \frac{\delta_k^2 L \|\mathbf{v}_{k+1} - \mathbf{x}_k\|^2}{2}$$

where (a) is because $\eta_k = \delta_k$. Then by the manner $\delta_k$ is chosen, we have

$$f(\mathbf{x}_{k+1}) - \Phi_{k+1}(\mathbf{v}_{k+1}) \tag{25}$$

$$= (1 - \delta_k)\big[f(\mathbf{x}_k) - \Phi_k(\mathbf{v}_k)\big] + \frac{\delta_k^2 L \|\mathbf{v}_{k+1} - \mathbf{x}_k\|^2}{2}$$

$$\overset{(b)}{\le} (1 - \tilde{\delta}_k)\big[f(\mathbf{x}_k) - \Phi_k(\mathbf{v}_k)\big] + \frac{\tilde{\delta}_k^2 L \|\mathbf{v}_{k+1} - \mathbf{x}_k\|^2}{2}$$

where in (b) $\tilde{\delta}_k \in [0,1]$. Choosing $\tilde{\delta}_k = 0$, we obtain $\mathcal{G}_{k+1} \le \mathcal{G}_k$. Choosing $\tilde{\delta}_k = \frac{2}{k+2}$, we obtain the convergence rate. $\qquad\square$

### B.5  Line search for Alg. 2

We can also choose the step size $\eta_k$ via line search, although this might be more computationally costly in practice because it requires computing the function value. The parameters are selected as

$$\delta_k = \frac{2}{k+2}, \ \forall k \ge 0 \tag{26a}$$

$$\eta_k = \underset{\eta \in [0,1]}{\arg\min} f\big((1 - \eta)\mathbf{x}_k + \eta\mathbf{v}_{k+1}\big). \tag{26b}$$

Such a parameter choice also ensures per step objective descent since

$$f(\mathbf{x}_{k+1}) = \min_{\eta \in [0,1]} f\big((1 - \eta)\mathbf{x}_k + \eta\mathbf{v}_{k+1}\big)$$

$$\overset{(a)}{\le} f\big((1 - \theta)\mathbf{x}_k + \theta\mathbf{v}_{k+1}\big) \overset{(b)}{=} f(\mathbf{x}_k)$$

where in (a) we have $\theta \in [0,1]$; and in (b) we set $\theta = 0$. Primal-dual convergence is established as follows.

**Theorem 6.** *If Assumptions 1-3 hold, while* $\delta_k$ *and* $\eta_k$ *are chosen via* (26), *Alg. 2 guarantees that*

$$\mathcal{G}_k = f(\mathbf{x}_k) - \Phi_k(\mathbf{v}_k) \le \frac{2LD^2}{k+1}, \ \forall k \ge 1.$$

*Proof.* Let $\tilde{\eta}_k = \frac{2}{k+2}, \forall k$. By the choice of $\eta_k$, we have

$$f(\mathbf{x}_{k+1}) = \min_{\eta \in [0,1]} f\big((1 - \eta)\mathbf{x}_k + \eta\mathbf{v}_{k+1}\big) \le f\big((1 - \tilde{\eta}_k)\mathbf{x}_k + \tilde{\eta}_k\mathbf{v}_{k+1}\big). \tag{27}$$

Then using smoothness, we arrive at

$$f(\mathbf{x}_{k+1}) - f(\mathbf{x}_k) \tag{28}$$
$$\leq f\big((1-\tilde{\eta}_k)\mathbf{x}_k + \tilde{\eta}_k\mathbf{v}_{k+1}\big) - f(\mathbf{x}_k)$$
$$\leq \tilde{\eta}_k\big\langle \nabla f(\mathbf{x}_k), \mathbf{v}_{k+1} - \mathbf{x}_k\big\rangle + \frac{\tilde{\eta}_k^2 L}{2}\|\mathbf{v}_{k+1} - \mathbf{x}_k\|^2.$$

Then combining (28) and (19), and following the same steps in (20), we can prove this theorem. $\square$

Through Theorem 6 it is straightforward to derive the primal and dual convergence, respectively, following the same argument of Corollary 1. For this reason, it is omitted here.

### B.6 Proof of Theorem 3

*Proof.* It can be seen that (21) still holds.

**Parameter-free step size.** Plugging in $\delta_k = \eta_k = \frac{1}{k+1}$ into (21), we arrive at

$$f(\mathbf{x}_{k+1}) - \Phi_{k+1}(\mathbf{v}_{k+1}) \leq (1-\delta_k)\big[f(\mathbf{x}_k) - \Phi_k(\mathbf{v}_k)\big] + \frac{\delta_k^2 L\|\mathbf{v}_{k+1} - \mathbf{x}_k\|^2}{2}$$

$$\leq \Delta(0,k)\big[f(\mathbf{x}_0) - \Phi_0(\mathbf{v}_0)\big] + \sum_{\tau=0}^{k}\frac{LD^2\delta_\tau^2}{2}\Delta(\tau+1,k)$$

$$= \mathcal{O}\Big(\frac{LD^2\ln(k+2)}{k+1}\Big) \tag{29}$$

where $\Delta(i,j) := \prod_{\tau=i}^{j}(1-\delta_\tau)$, $\Phi_0(\mathbf{x}) \equiv 0$, and $\mathbf{v}_0$ is any point in $\mathcal{X}$.

**Smooth step size.** Notice that the choice of $\eta_k$ minimizes the RHS of (21) when $\delta_k$ is fixed, then we have

$$f(\mathbf{x}_{k+1}) - \Phi_{k+1}(\mathbf{v}_{k+1}) \tag{30}$$

$$\leq (1-\delta_k)\big[f(\mathbf{x}_k) - \Phi_k(\mathbf{v}_k)\big] + (\eta_k - \delta_k)\big\langle \nabla f(\mathbf{x}_k), \mathbf{v}_{k+1} - \mathbf{x}_k\big\rangle + \frac{\eta_k^2 L\|\mathbf{v}_{k+1} - \mathbf{x}_k\|^2}{2}$$

$$\overset{(a)}{\leq} (1-\delta_k)\big[f(\mathbf{x}_k) - \Phi_k(\mathbf{v}_k)\big] + (\tilde{\eta}_k - \delta_k)\big\langle \nabla f(\mathbf{x}_k), \mathbf{v}_{k+1} - \mathbf{x}_k\big\rangle + \frac{\tilde{\eta}_k^2 L\|\mathbf{v}_{k+1} - \mathbf{x}_k\|^2}{2}$$

$$\overset{(b)}{\leq} (1-\delta_k)\big[f(\mathbf{x}_k) - \Phi_k(\mathbf{v}_k)\big] + \frac{\delta_k^2 L\|\mathbf{v}_{k+1} - \mathbf{x}_k\|^2}{2}$$

$$= \mathcal{O}\Big(\frac{LD^2\ln(k+2)}{k+1}\Big)$$

where in (a) $\tilde{\eta}_k \in [0,1]$; and in (b) we set $\tilde{\eta}_k = \delta_k$.

**Line search.** When $\eta_k$ is chosen via line search, we have for any $\tilde{\eta}_k \in [0,1]$

$$f(\mathbf{x}_{k+1}) = \min_{\eta \in [0,1]} f\big((1-\eta)\mathbf{x}_k + \eta\mathbf{v}_{k+1}\big) \leq f\big((1-\tilde{\eta}_k)\mathbf{x}_k + \tilde{\eta}_k\mathbf{v}_{k+1}\big). \tag{31}$$

Then by smoothness, we have

$$f(\mathbf{x}_{k+1}) - f(\mathbf{x}_k) \leq f\big((1-\tilde{\eta}_k)\mathbf{x}_k + \tilde{\eta}_k\mathbf{v}_{k+1}\big) - f(\mathbf{x}_k) \tag{32}$$

$$\leq \tilde{\eta}_k\big\langle \nabla f(\mathbf{x}_k), \mathbf{v}_{k+1} - \mathbf{x}_k\big\rangle + \frac{\tilde{\eta}_k^2 L}{2}\|\mathbf{v}_{k+1} - \mathbf{x}_k\|^2.$$

Then using the same argument as the derivation of (21), we can obtain

$$f(\mathbf{x}_{k+1}) - \Phi_{k+1}(\mathbf{v}_{k+1}) \tag{33}$$

$$\leq (1-\delta_k)\big[f(\mathbf{x}_k) - \Phi_k(\mathbf{v}_k)\big] + (\tilde{\eta}_k - \delta_k)\big\langle \nabla f(\mathbf{x}_k), \mathbf{v}_{k+1} - \mathbf{x}_k\big\rangle + \frac{\tilde{\eta}_k^2 L\|\mathbf{v}_{k+1} - \mathbf{x}_k\|^2}{2}.$$

Simply setting $\tilde{\eta}_k = \frac{1}{k+1}$, and using the same derivation as in (30), the proof can be completed. $\square$

**B.7 Proof for choosing $\delta_k = \delta$**

When Assumptions 1 is satisfied w.r.t. $\ell_2$-norm, we show the following parameter choice in Alg. 2 leads to convergence as well.

$$\delta_k = \delta, \; \eta_k = \frac{c}{k + k_0}, \; \forall k \geq 0 \tag{34}$$

where $\delta \in (0, 1)$, and $c$ and $k_0$ are constants to be specified later. Due to the choice of $\delta_k = \delta$, $\mathbf{g}_{k+1}$ is an exponentially moving average of previous gradients. Note that the moving average was adopted in [27] for stochastic FW to reduce the mean square error of the noisy gradient. However, we use it in a totally different purpose.

**Lemma 2.** *Choose parameters as in* (34). *Suppose there exist a constant $c_0$ that satisfies*

$$c_1^2 \leq \left[ 1 - (1 - \delta) \frac{(k_0 + 1)^2}{k_0^2} \right] \delta c_0^2 \tag{35}$$

*then it is guaranteed that*

$$\|\mathbf{g}_{k+1} - \nabla f(\mathbf{x}_k)\|_2^2 \leq \frac{c_0^2 L^2 D^2}{(k + k_0)^2}.$$

*Proof.*

$$\|\mathbf{g}_{k+1} - \nabla f(\mathbf{x}_k)\|_2^2 \tag{36}$$

$$= (1 - \delta)^2 \|\mathbf{g}_k - \nabla f(\mathbf{x}_k)\|_2^2$$

$$= (1 - \delta)^2 \|\mathbf{g}_k - \nabla f(\mathbf{x}_{k-1}) + \nabla f(\mathbf{x}_{k-1}) - \nabla f(\mathbf{x}_k)\|_2^2$$

$$\overset{(a)}{\leq} (1 - \delta)^2 (1 + \theta) \|\mathbf{g}_k - \nabla f(\mathbf{x}_{k-1})\|_2^2 + (1 - \delta)^2 (1 + \frac{1}{\theta}) \|\nabla f(\mathbf{x}_{k-1}) - \nabla f(\mathbf{x}_k)\|_2^2$$

$$\overset{(b)}{\leq} (1 - \delta)^2 (1 + \theta) \|\mathbf{g}_k - \nabla f(\mathbf{x}_{k-1})\|_2^2 + (1 - \delta)^2 (1 + \frac{1}{\theta}) L^2 \eta_{k-1}^2 \|\mathbf{x}_{k-1} - \mathbf{v}_k\|_2^2$$

$$\overset{(c)}{\leq} (1 - \delta)^2 (1 + \theta) \|\mathbf{g}_k - \nabla f(\mathbf{x}_{k-1})\|_2^2 + (1 - \delta)^2 (1 + \frac{1}{\theta}) L^2 D^2 \eta_{k-1}^2$$

$$\overset{(d)}{\leq} (1 - \delta) \|\mathbf{g}_k - \nabla f(\mathbf{x}_{k-1})\|_2^2 + (1 - \delta)^2 (1 + \frac{1}{\delta}) L^2 D^2 \eta_{k-1}^2$$

$$\overset{(e)}{\leq} (1 - \delta) \|\mathbf{g}_k - \nabla f(\mathbf{x}_{k-1})\|_2^2 + L^2 D^2 \frac{\eta_{k-1}^2}{\delta}$$

where (a) is by Young's inequality with $\theta > 0$ to be specified later; (b) follows from Assumption 1; (c) is because Assumption 3; in (d) we choose $\theta = \delta < 1$ and use the fact that $(1 - \delta)^2 (1 + \delta) \leq (1 - \delta)$; and (e) uses $\delta \leq 1$ so that $(1 - \delta)^2 (1 + \frac{1}{\delta}) = \frac{1}{\delta} - 1 + \delta^2 - 2\delta \leq \frac{1}{\delta}$.

We proof this lemma by induction. Given the choice of $\mathbf{g}_0 = \nabla f(\mathbf{x}_0)$, we must have $\mathbf{g}_1 = \nabla f(\mathbf{x}_0)$, which implies $\|\mathbf{g}_1 - \nabla f(\mathbf{x}_0)\|_2^2 = 0 \leq \frac{c_0^2 L^2 D^2}{k_0^2}$ directly. Next we assume that $\|\mathbf{g}_k - \nabla f(\mathbf{x}_{k-1})\|_2^2 \leq \frac{c_0^2 L^2 D^2}{(k-1+k_0)^2}$ holds for some $k \geq 1$. Using (36), we have

$$\|\mathbf{g}_{k+1} - \nabla f(\mathbf{x}_k)\|_2^2 \leq (1 - \delta) \|\mathbf{g}_k - \nabla f(\mathbf{x}_{k-1})\|_2^2 + L^2 D^2 \frac{\eta_{k-1}^2}{\delta}$$

$$\leq (1 - \delta) \frac{c_0^2 L^2 D^2}{(k + k_0 - 1)^2} + L^2 D^2 \frac{\eta_{k-1}^2}{\delta}$$

$$\leq (1 - \delta) \frac{c_0^2 L^2 D^2}{(k + k_0 - 1)^2} + L^2 D^2 \frac{c_1^2}{\delta(k + k_0)^2}$$

$$= (1 - \delta) \frac{c_0^2 L^2 D^2}{(k + k_0)^2} \frac{(k + k_0)^2}{(k + k_0 - 1)^2} + L^2 D^2 \frac{c_1^2}{\delta(k + k_0)^2}$$

$$\leq (1 - \delta) \frac{c_0^2 L^2 D^2}{(k + k_0)^2} \frac{(k_0 + 1)^2}{k_0^2} + L^2 D^2 \frac{c_1^2}{\delta(k + k_0)^2}$$

$$\leq \frac{c_0^2 L^2 D^2}{(k + k_0)^2} \tag{37}$$

where the last inequality comes from the choice of $c_1$. The proof is thus completed. $\qquad\square$

To avoid the complexity of choosing constants, we consider an instance where $k_0 = 2$, $\delta = 0.8$, $c_1 = 2$, and $c_0 \approx 3.05$. It can be verified that (35) is satisfied. Then applying Lemma 2, the convergence of Alg.2 can be obtained.

**Theorem 7.** *Let* $\mathbf{g}_0 = \nabla f(\mathbf{x}_0)$, $\eta_k = \frac{2}{k+3}$, *and* $\delta = 0.8$. *Then for* $\forall k \geq 1$, *the convergence rate of Alg. 2 with* (34) *is*

$$f(\mathbf{x}_k) - f(\mathbf{x}^*) = \mathcal{O}\Big(\frac{LD^2}{k}\Big).$$

*Proof.* Using Assumption 1, we have

$$f(\mathbf{x}_{k+1}) - f(\mathbf{x}^*) \leq f(\mathbf{x}_k) - f(\mathbf{x}^*) + \big\langle \nabla f(\mathbf{x}_k), \mathbf{x}_{k+1} - \mathbf{x}_k \big\rangle + \frac{L}{2}\|\mathbf{x}_{k+1} - \mathbf{x}_k\|_2^2 \qquad (38)$$

$$= f(\mathbf{x}_k) - f(\mathbf{x}^*) + \eta_k\big\langle \nabla f(\mathbf{x}_k), \mathbf{v}_{k+1} - \mathbf{x}_k \big\rangle + \frac{\eta_k^2 L}{2}\|\mathbf{v}_{k+1} - \mathbf{x}_k\|_2^2$$

$$\leq f(\mathbf{x}_k) - f(\mathbf{x}^*) + \eta_k\big\langle \nabla f(\mathbf{x}_k), \mathbf{v}_{k+1} - \mathbf{x}_k \big\rangle + \frac{\eta_k^2 LD^2}{2}.$$

Next we have

$$\big\langle \nabla f(\mathbf{x}_k), \mathbf{v}_{k+1} - \mathbf{x}_k \big\rangle = \big\langle \nabla f(\mathbf{x}_k), \mathbf{x}^* - \mathbf{x}_k \big\rangle + \big\langle \nabla f(\mathbf{x}_k), \mathbf{v}_{k+1} - \mathbf{x}^* \big\rangle$$

$$\overset{(a)}{\leq} f(\mathbf{x}^*) - f(\mathbf{x}_k) + \big\langle \nabla f(\mathbf{x}_k), \mathbf{v}_{k+1} - \mathbf{x}^* \big\rangle$$

$$= f(\mathbf{x}^*) - f(\mathbf{x}_k) + \big\langle \mathbf{g}_{k+1}, \mathbf{v}_{k+1} - \mathbf{x}^* \big\rangle + \big\langle \nabla f(\mathbf{x}_k) - \mathbf{g}_{k+1}, \mathbf{v}_{k+1} - \mathbf{x}^* \big\rangle$$

$$\overset{(b)}{\leq} f(\mathbf{x}^*) - f(\mathbf{x}_k) + \big\langle \nabla f(\mathbf{x}_k) - \mathbf{g}_{k+1}, \mathbf{v}_{k+1} - \mathbf{x}^* \big\rangle$$

$$\leq f(\mathbf{x}^*) - f(\mathbf{x}_k) + D\|\nabla f(\mathbf{x}_k) - \mathbf{g}_{k+1}\|_2 \qquad (39)$$

where (a) is by the convexity of $f(\mathbf{x})$; (b) is because $\mathbf{v}_{k+1}$ minimizes $\langle \mathbf{g}_{k+1}, \mathbf{x} \rangle$ over $\mathcal{X}$; and the last inequality relies on Cauchy-Schwarz inequality and Assumption 3. Plugging (39) into (38), we have

$$f(\mathbf{x}_{k+1}) - f(\mathbf{x}^*) \leq (1 - \eta_k)\big[f(\mathbf{x}_k) - f(\mathbf{x}^*)\big] + \eta_k D\|\nabla f(\mathbf{x}_k) - \mathbf{g}_{k+1}\|_2 + \frac{\eta_k^2 LD^2}{2}. \qquad (40)$$

Let $\xi_k = \frac{\eta_k c_0 LD^2}{k+k_0} + \frac{\eta_k^2 LD^2}{2}$, then we have

$$f(\mathbf{x}_{k+1}) - f(\mathbf{x}^*) \leq (1 - \eta_k)\big[f(\mathbf{x}_k) - f(\mathbf{x}^*)\big] + \eta_k D\|\nabla f(\mathbf{x}_k) - \mathbf{g}_{k+1}\|_2 + \frac{\eta_k^2 LD^2}{2}$$

$$\leq (1 - \eta_k)\big[f(\mathbf{x}_k) - f(\mathbf{x}^*)\big] + \xi_k$$

$$= \big[f(\mathbf{x}_0) - f(\mathbf{x}^*)\big] \prod_{\tau=0}^{k}(1 - \eta_\tau) + \sum_{\tau=0}^{k} \xi_\tau \prod_{j=\tau+1}^{k}(1 - \eta_j)$$

$$= \mathcal{O}\Big(\frac{LD^2}{k}\Big). \qquad (41)$$

The proof is thus completed. $\qquad\square$

### B.8  Additional discussions

Many of existing works, e.g., [14], study (projected) heavy ball momentum by introducing auxiliary variables $\mathbf{z}_k$ such that the update on variable $\mathbf{x}_k$ can be viewed as a "gradient update" on $\mathbf{z}_k$, i.e., $\mathbf{z}_{k+1} = \mathbf{z}_k - \eta \nabla f(\mathbf{x}_k)$. By constructing the $\{\mathbf{z}_k\}$ sequence, it is possible to view heavy ball momentum approximately as GD. Though this trick is smart and analytically convenient, it does not give too much insight for the heavy ball momentum itself.

By comparing the use of heavy ball momentum in FW and GD, it may suggest new perspectives. For example, one can view Alg.2 as the dual-averaging version of FW as well. This suggests that it is intriguing to study (projected) heavy ball momentum from dual-averaging point of view. This is slightly off the main theme of this work, and we leave it for future research.

## C  Stopping criterion

Recall that for a prescribed $\epsilon > 0$, having $f(\mathbf{x}_k) - \Phi_k(\mathbf{v}_k) \leq \epsilon$ directly implies $f(\mathbf{x}_k) - f(\mathbf{x}^*) \leq \epsilon$. Next, we show how to update $\Phi_k(\mathbf{v}_k)$ iteratively in order to obtain a stopping criterion. Let us note that

$$\Phi_{k+1}(\mathbf{x}) = \sum_{\tau=0}^{k} w_k^{\tau} \big[ f(\mathbf{x}_\tau) + \langle \nabla f(\mathbf{x}_\tau), \mathbf{x} - \mathbf{x}_\tau \rangle \big]$$

$$= \sum_{\tau=0}^{k} w_k^{\tau} \big[ f(\mathbf{x}_\tau) - \langle \nabla f(\mathbf{x}_\tau), \mathbf{x}_\tau \rangle \big] + \langle \mathbf{g}_{k+1}, \mathbf{x} \rangle$$

$$:= C_{k+1} + \langle \mathbf{g}_{k+1}, \mathbf{x} \rangle, \ \forall k \geq 0.$$

Hence, to compute $\Phi_{k+1}(\mathbf{v}_{k+1})$, we only need to update $C_{k+1}$ iteratively. A simple derivation leads to

$$C_{k+1} = (1 - \delta_k) C_k + \delta_k \Big[ f(\mathbf{x}_k) - \langle \nabla f(\mathbf{x}_k), \mathbf{x}_k \rangle \Big],$$
$$\text{with } C_1 = f(\mathbf{x}_0) - \langle \nabla f(\mathbf{x}_0), \mathbf{x}_0 \rangle. \tag{42}$$

In sum, one can efficiently obtain $\Phi_{k+1}(\mathbf{v}_{k+1})$ as

$$\Phi_{k+1}(\mathbf{v}_{k+1}) = C_{k+1} + \langle \mathbf{g}_{k+1}, \mathbf{v}_{k+1} \rangle \tag{43}$$

with $C_{k+1}$ recursively updated via (42).

## D  Missing proofs in Section 4

### D.1  Proof of Theorem 4

*Proof.* Consider the case where $\eta_k^s = \delta_k^s$. Using Assumption 1, we have

$$f(\mathbf{x}_{k+1}^s) - f(\mathbf{x}_k^s) \leq \langle \nabla f(\mathbf{x}_k^s), \mathbf{x}_{k+1}^s - \mathbf{x}_k^s \rangle + \frac{L}{2} \|\mathbf{x}_{k+1}^s - \mathbf{x}_k^s\|^2 \tag{44}$$

$$= \eta_k^s \langle \nabla f(\mathbf{x}_k^s), \mathbf{v}_{k+1}^s - \mathbf{x}_k^s \rangle + \frac{(\eta_k^s)^2 L}{2} \|\mathbf{v}_{k+1}^s - \mathbf{x}_k^s\|^2.$$

Then we have

$$\Phi_{k+1}^s(\mathbf{v}_{k+1}^s) = (1 - \delta_k^s) \Phi_k^s(\mathbf{v}_{k+1}^s) + \delta_k^s \Big[ f(\mathbf{x}_k^s) + \langle \nabla f(\mathbf{x}_k^s), \mathbf{v}_{k+1}^s - \mathbf{x}_k^s \rangle \Big] \tag{45}$$

$$\geq (1 - \delta_k^s) \Phi_k^s(\mathbf{v}_k^s) + \delta_k^s \Big[ f(\mathbf{x}_k^s) + \langle \nabla f(\mathbf{x}_k^s), \mathbf{v}_{k+1}^s - \mathbf{x}_k^s \rangle \Big].$$

Now subtracting $\Phi_{k+1}^s(\mathbf{v}_{k+1}^s)$ on both sides of (44), we have

$$f(\mathbf{x}_{k+1}^s) - \Phi_{k+1}^s(\mathbf{v}_{k+1}^s) \tag{46}$$

$$\leq f(\mathbf{x}_k^s) + \eta_k^s \langle \nabla f(\mathbf{x}_k^s), \mathbf{v}_{k+1}^s - \mathbf{x}_k^s \rangle + \frac{(\eta_k^s)^2 L \|\mathbf{v}_{k+1}^s - \mathbf{x}_k^s\|^2}{2} - \Phi_{k+1}^s(\mathbf{v}_{k+1}^s)$$

$$\overset{(a)}{\leq} (1 - \delta_k^s) \big[ f(\mathbf{x}_k^s) - \Phi_k^s(\mathbf{v}_k^s) \big] + \frac{(\delta_k^s)^2 L \|\mathbf{v}_{k+1}^s - \mathbf{x}_k^s\|^2}{2}$$

$$\overset{(b)}{\leq} (1 - \delta_k^s) \big[ f(\mathbf{x}_k^s) - \Phi_k^s(\mathbf{v}_k^s) \big] + \frac{(\delta_k^s)^2 L D^2}{2}$$

where (a) uses $\eta_k^s = \delta_k^s$ and (45); and (b) relies on Assumption 3. For convenience, let us define $\Delta^s(i,j) := \prod_{\tau=i}^{j} (1 - \delta_\tau^s)$. Then unrolling (46), we get

$$f(\mathbf{x}_{k+1}^s) - \Phi_{k+1}^s(\mathbf{v}_{k+1}^s)$$

$$\leq \Delta^s(0,k) \big[ f(\mathbf{x}_0^s) - \Phi_0^s(\mathbf{v}_0^s) \big] + \sum_{\tau=0}^{k} \frac{L D^2 (\delta_\tau^s)^2}{2} \Delta^s(\tau+1, k)$$

$$\leq \frac{C^s(C^s+1)}{(k+1+C^s)(k+2+C^s)} \big[ f(\mathbf{x}_0^s) - \Phi_0^s(\mathbf{v}_0^s) \big] + \frac{2(k+1)L D^2}{(k+1+C^s)(k+2+C^s)}.$$

When $s = 0$, plugging $C^0 = 0$, we have

$$f(\mathbf{x}_{k+1}^0) - \Phi_{k+1}(\mathbf{v}_{k+1}^0) \leq \frac{2LD^2}{k+2}. \tag{47}$$

Hence (14a) in Theorem 4 is proved. Next consider $s \geq 1$. Using the observation that $f(\mathbf{x}_0^s) - \Phi_0^s(\mathbf{v}_0^s) = \bar{\mathcal{G}}_{K_{s-1}}^{s-1} < \mathcal{G}_{K_{s-1}}^{s-1}$, we then have

$$\mathcal{G}_{k+1}^s = f(\mathbf{x}_{k+1}^s) - \Phi_{k+1}^s(\mathbf{v}_{k+1}^s) \tag{48}$$

$$< \frac{C^s(C^s+1)}{(k+1+C^s)(k+2+C^s)}\mathcal{G}_{K_{s-1}}^{s-1} + \frac{2(k+1)LD^2}{(k+1+C^s)(k+2+C^s)}$$

$$\stackrel{(c)}{=} \frac{2LD^2(C^s+1)}{(k+1+C^s)(k+2+C^s)} + \frac{2(k+1)LD^2}{(k+1+C^s)(k+2+C^s)} = \frac{2LD^2}{k+1+C^s}.$$

where (c) uses the definition of $C^s$. Hence (14b) in Theorem 4 is proved.

Finally, we only need to show that $C^s \geq 1 + \sum_{j=0}^{s-1} K_j$ by induction. First by definition of $C^1 = 2LD^2/(\mathcal{G}_{K_0}^0)$, with $\mathcal{G}_{K_0}^0 \leq \frac{2LD^2}{K_0+1}$, it is clear that $C^1 \geq 1 + K_0$. Then suppose $C^s \geq 1 + \sum_{j=0}^{s-1} K_j$ hold for some $s$, we will show that $C^{s+1} \geq 1 + \sum_{j=0}^{s} K_j$.

Using (48), we have $C^{s+1} = 2LD^2/(\mathcal{G}_{K_s}^s) \geq C^s + K_s \geq 1 + \sum_{j=0}^{s-1} K_j + K_s$. Hence (14b) is proved.

For the smooth step size (12) and line search (13), the same bound can be obtained by using the same arguments as in Theorems 2 and 6. Hence they are omitted here. $\qquad\square$

## E  Directionally smooth step size

### E.1  Proof of Corollary 2

*Proof.* Using Definition 2 and following the standard derivation of descent lemma [29, Lemma 1.2.3], we can show that

$$f(\mathbf{x}_{k+1}) - f(\mathbf{x}_k) \tag{49}$$

$$\leq \eta_k\langle\nabla f(\mathbf{x}_k), \mathbf{v}_{k+1} - \mathbf{x}_k\rangle + \frac{\eta_k^2 L(\mathbf{x}_k, \mathbf{x}_{k+1})}{2}\|\mathbf{v}_{k+1} - \mathbf{x}_k\|^2$$

$$\leq \eta_k\langle\nabla f(\mathbf{x}_k), \mathbf{v}_{k+1} - \mathbf{x}_k\rangle + \frac{\eta_k^2 L(\mathbf{x}_k, \mathbf{v}_{k+1})}{2}\|\mathbf{v}_{k+1} - \mathbf{x}_k\|^2.$$

The reason for $L(\mathbf{x}_k, \mathbf{v}_{k+1}) \geq L(\mathbf{x}_k, \mathbf{x}_{k+1})$ is that $\mathbf{x}_{k+1}$ lives in between $\mathbf{x}_k$ and $\mathbf{v}_{k+1}$. Although $L(\mathbf{x}_k, \mathbf{x}_{k+1})$ can provide a tighter bound, it is not tractable.

Combining (49) and (19), we have

$$f(\mathbf{x}_{k+1}) - \Phi_{k+1}(\mathbf{v}_{k+1}) \tag{50}$$

$$\leq (1-\delta_k)\big[f(\mathbf{x}_k) - \Phi_k(\mathbf{v}_k)\big] + (\eta_k - \delta_k)\langle\nabla f(\mathbf{x}_k), \mathbf{v}_{k+1} - \mathbf{x}_k\rangle + \frac{\eta_k^2 L(\mathbf{x}_k, \mathbf{v}_{k+1})\|\mathbf{v}_{k+1} - \mathbf{x}_k\|^2}{2}.$$

It can be verified that the specific choice of $\eta_k$ in (10) minimizes the RHS of (50) over $[0, 1]$. Hence we have

$$f(\mathbf{x}_{k+1}) - \Phi_{k+1}(\mathbf{v}_{k+1}) \tag{51}$$

$$\leq (1-\delta_k)\big[f(\mathbf{x}_k) - \Phi_k(\mathbf{v}_k)\big] + \frac{\eta_k^2 L(\mathbf{x}_k, \mathbf{v}_{k+1})\|\mathbf{v}_{k+1} - \mathbf{x}_k\|^2}{2} + (\eta_k - \delta_k)\langle\nabla f(\mathbf{x}_k), \mathbf{v}_{k+1} - \mathbf{x}_k\rangle$$

$$\stackrel{(a)}{\leq} (1-\delta_k)\big[f(\mathbf{x}_k) - \Phi_k(\mathbf{v}_k)\big] + \frac{\alpha_k^2 L(\mathbf{x}_k, \mathbf{v}_{k+1})\|\mathbf{v}_{k+1} - \mathbf{x}_k\|^2}{2} + (\alpha_k - \delta_k)\langle\nabla f(\mathbf{x}_k), \mathbf{v}_{k+1} - \mathbf{x}_k\rangle$$

$$\stackrel{(b)}{=} (1-\delta_k)\big[f(\mathbf{x}_k) - \Phi_k(\mathbf{v}_k)\big] + \frac{\delta_k^2 L(\mathbf{x}_k, \mathbf{v}_{k+1})\|\mathbf{v}_{k+1} - \mathbf{x}_k\|^2}{2}$$

$$\stackrel{(c)}{=} (1-\delta_k)\big[f(\mathbf{x}_k) - \Phi_k(\mathbf{v}_k)\big] + \frac{\delta_k^2 L\|\mathbf{v}_{k+1} - \mathbf{x}_k\|^2}{2}$$

$$\leq \frac{2LD^2}{k+2}$$

where in (a) $\alpha_k$ can be chosen as any number in $[0, 1]$; in (b) we set $\alpha_k = \delta_k$; and (c) uses $L(\mathbf{x}_k, \mathbf{v}_{k+1}) \leq L$. This completes the proof. $\qquad\square$

## E.2 Computing directionally smooth constant

Define a one dimensional function $g(\eta) := f\big(\mathbf{x}_k + \eta(\mathbf{v}_{k+1} - \mathbf{x}_k)\big)$, where $\mathrm{dom}\,\eta = [0, 1]$. Then it is clear that $\nabla g(\eta) = \langle \mathbf{v}_{k+1} - \mathbf{x}_k, \nabla f\big(\mathbf{x}_k + \eta(\mathbf{v}_{k+1} - \mathbf{x}_k)\big)\rangle$. Therefore, it is easy to see that $g(\eta)$ is smooth, i.e.,

$$
\begin{aligned}
\big|\nabla g(\eta_1) - \nabla g(\eta_2)\big| &= |\langle \mathbf{v}_{k+1} - \mathbf{x}_k, \nabla f\big(\mathbf{x}_k + \eta_1(\mathbf{v}_{k+1} - \mathbf{x}_k)\big) - \nabla f\big(\mathbf{x}_k + \eta_2(\mathbf{v}_{k+1} - \mathbf{x}_k)\big)\rangle| \\
&\leq \|\mathbf{v}_{k+1} - \mathbf{x}_k\|\big\|\nabla f\big(\mathbf{x}_k + \eta_1(\mathbf{v}_{k+1} - \mathbf{x}_k)\big) - \nabla f\big(\mathbf{x}_k + \eta_2(\mathbf{v}_{k+1} - \mathbf{x}_k)\big)\big\|_* \\
&\leq L(\mathbf{x}_k, \mathbf{v}_{k+1})\|\mathbf{v}_{k+1} - \mathbf{x}_k\|^2|\eta_1 - \eta_2| \qquad\qquad (52)
\end{aligned}
$$

On the other hand, one can also analytically find $L_g$ by definition; i.e., $\big|\nabla g(\eta_1) - \nabla g(\eta_2)\big| \leq L_g\big|\eta_1 - \eta_2\big|$. Comparing $L_g$ with RHS of (52), we can obtain $L(\mathbf{x}_k, \mathbf{v}_{k+1})$. This method can be applied when $f$ is e.g., quadratic loss and logistic loss.

# F  More on numerical tests

All numerical experiments are performed using Python 3.7 on an Intel i7-4790CPU @3.60 GHz (32 GB RAM) desktop.

## F.1  Binary classification

Table 2: A summary of datasets used in numerical tests

| Dataset | $d$ | $N$ (train) | nonzeros |
|---------|------|-------------|----------|
| *w7a* | 300 | $24,692$ | $3.89\%$ |
| *realsim* | $20,958$ | $50,617$ | $0.24\%$ |
| *mushromm* | 122 | $8,124$ | $18.75\%$ |
| *ijcnn1* | 22 | $49,990$ | $40.91\%$ |

**Sparsity promoting property of FW variants for $\ell_1$-norm ball constraint.** FW in Alg. 1 directly promotes sparsity on the solution if it is initialized at $\mathbf{x}_0 = \mathbf{0}$. To see this, suppose that the $i$-th entry of $\nabla f(\mathbf{x}_k)$ has the largest absolute value, then we have $\mathbf{v}_{k+1} = [0, \ldots, -\mathrm{sgn}\big([\nabla f(\mathbf{x}_k)]_i\big)R, \ldots, 0]^\top$ with the $i$-th entry being non-zero. Hence, $\mathbf{x}_k$ has at most $k$ non-zero entries given $k-1$ entries are non-zero in $\mathbf{x}_{k-1}$. This sparsity promoting property also holds for Alg. 2 for the same reason.

## F.2  Matrix completion

The dataset used for the test is *MovieLens100K*, where 1682 movies are rated by 943 users with $6.30\%$ ratings observed. The initialization and data processing are the same as those used in [11].

Besides the projection-free property, FW and its variants are more suitable for problem (16) compared to GD because they also guarantee $\mathrm{rank}(\mathbf{X}_k) \leq k + 1$ [11, 15]. Take FW in Alg. 1 for example. First it is clear that $\nabla f(\mathbf{X}_k) = (\mathbf{X}_k - \mathbf{A})_\mathcal{K}$. Suppose that the SVD of $\nabla f(\mathbf{X}_k)$ is given by $\nabla f(\mathbf{X}_k) = \mathbf{P}_k\boldsymbol{\Sigma}_k\mathbf{Q}_k^\top$. Then the FW subproblem can be solved easily by

$$
\boldsymbol{V}_{k+1} = -R\mathbf{p}_k\mathbf{q}_k^\top \qquad\qquad (53)
$$

where $\mathbf{p}_k$ and $\mathbf{q}_k$ denote the left and right singular vectors corresponding to the largest singular value of $\nabla f(\mathbf{X}_k)$, respectively. Clearly $\boldsymbol{V}_{k+1}$ in (53) has rank at most 1. Hence it is easy to see $\mathbf{X}_{k+1} = (1 - \delta_k)\mathbf{X}_k + \delta_k\boldsymbol{V}_{k+1}$ has rank at most $k + 2$ if $\mathbf{X}_k$ is a rank-$(k + 1)$ matrix (i.e., $\mathbf{X}_0$ has rank 1). Using similar arguments, Alg. 2 also ensures $\mathrm{rank}(\mathbf{X}_k) \leq k + 1$. Therefore, the low rank structure is directly promoted by FW variants.