# OpenReview forum: "Heavy Ball Momentum for Conditional Gradient"
_NeurIPS.cc/2021/Conference — NeurIPS 2021 Poster_

### Official Review · Reviewer_rTkg · 2021-07-10

**Rating:** 7
**Confidence:** 4

**Summary:**

This paper proposes a conditional gradient algorithm that uses heavy ball type acceleration to solve the constrained optimization problem. After introducing the vanilla Frank Wolfe method which is one of the common methods to solve these types of problems, authors introduce their FW with heavy ball momentum and provide convergence guarantees on L-smooth and convex objectives under various step-size choices. Furthermore, they also consider another version of the algorithm (FW with restart) which solves two subproblems to further improve the primal-dual error in the problem. Lastly, the authors present the performance of their algorithms on binary classification and matrix completion.



**Ethical Concerns:**

I don't see any ethical issues with this paper.

**Limitations And Societal Impact:**

Fairness in machine learning is becoming more important nowadays and there has been some work to provide more fair (!) models in machine learning. One of the proposed methods is to introduce some fair constraints (!) to optimization problems (e.g. Fairness Constraints: A Flexible Approach for Fair Classification by Zafar et al). Therefore, I believe this work can have potential in the context of Fairness in ML.

**Main Review:**

I believe heavy ball adaptation of the FW algorithm is an original idea. In practice, computing Lipschitz constants may be expensive or even impossible; but these assumptions are common in the literature and provides helpful tools to understand the performance of the algorithms. In that perspective, I believe the authors did a good work providing convergence analysis for various step-size choices (especially the parameter-free step size which does not require computation of these parameters).

I am concerned about the computation cost of solving two subproblems at Algorithm 3, even though restart improves the PD error. Can authors also add the computation time or memory use comparisons of these algorithms? I suspect that the restart approach may be costly at large problems and the benefit we get from restart may not compensate for the PD error improvement.

I found the discussions in the paper clear; however, it would be great if the authors can provide the advantage of heavy ball momentum over Nesterov's momentum numerically. In addition to discussion on the paper, including a comparison of memory use (or Type II PD error) on these algorithms could show the advantage of HFW over Nesterov or make the motivation more clear.

Lastly, best to my knowledge, projected gradient descent methods such as projected GD are commonly used in practice. Therefore, including projected GD in the numerical experiments (in addition to FW variants) would help to understand the performance of the algorithm.

**Time Spent Reviewing:**

2

---

> ### Author Response · Authors · 2021-08-08
> **Response to Reviewer rTkg**
>
> Thanks for the nice suggestions. More numerical tests will be added in the revised version.
>
> Q1: Should one be concerned about the “two subproblems” in Algorithm 3?
>
> A1: Algorithm 3 is mainly of theoretical interest as it offers deeper understanding of the generalized FW gap, along with a tighter bound. On the other hand, HFW is more practical thanks to its simplicity. Regarding numerical results, we tested Algorithm 3 on the l1 norm ball using the `mushroom’ dataset. Algorithm 3 slightly improves over HFW(WFW) when plotting iteration versus primal error. When comparing runtime, HFW performs better since Algorithm 3 roughly doubles the computational burden.
>
> Q2: How about additional numerical tests on Nesterov and projected GD?
>
> A2: We will add more numerical results in the updated version, as suggested. But for certain constraint sets, such as the nuclear norm balls, projection can be costly. In addition, there are no efficient methods available for projection on to the $n$-support norm balls yet. This amplifies the merits of FW type algorithms.
>
> Q3: How about social impact?
>
> A3: Thank you for pointing out the potential this work can have on fairness in ML. This is a nice direction to pursue, and we have included it in our future research agenda.

---

### Official Review · Reviewer_DsMv · 2021-07-15

**Rating:** 7
**Confidence:** 4

**Summary:**

This paper considers the effect of the heavy ball momentum on the Frank-Wolfe method. The $O(1/k)$ last-term convergence (type II as described in the paper) of a generalized FW gap is provided as the main contribution of this paper. Such a result is missing in the previous literature without the extra computational burden (only $O(1/\sqrt{k})$ result is available). The extension of the FW gap to the proposed generalized FW gap is natural, and this proposed quantity can also serve as the stopping criterion in practice.
Other than the classic step-size choice of $2/(k+2)$, a similar result is obtained under smooth step size with the additional beneﬁt of a non-increasing objective value, but at the cost of some prior knowledge of the smoothness parameter $L$. The problem of the pessimistic estimation of $L$ is then alleviated with the introduction of the directionally smooth step size.
Finally, the authors show that the restart technique further improves the last-term convergence.

**Limitations And Societal Impact:**

This work is theoretical and I believe there is no potential negative societal impact.

**Main Review:**

Originality and quality:
The result is interesting to me as the momentum technique has been the "crown jewel" of first order optimization but is rarely studied in the literature of Frank-Wolfe. The slowness of the FW is usually attributed to the zig-zag phenomenon and "smoothing" the update direction seems (and is proved to be in this paper) a plausible strategy to alleviate this issue.

Clarity:
This paper is well written and I enjoy reading it (there could be some bias because I am familiar with the literature already). The idea of generalized FW gap is well explained.

Significance:
This is an important result to the literature of Frank-Wolfe method.

Questions:
In the experiment section, the first row of Figure 1 looks impressive. However, since the constraint is the `$\ell$2-norm ball, the HFW method is equivalent to the normalized gradient descent method with heavy ball momentum (with an additional moving average on the variable). The zig-zag phenomenon is potentially missing in this setting and such a success may be due to that the HFW in this setting is indeed of the $O(1/k^2)$ accelerated convergence rate. The authors should clarify this point in their revision because this will not be a common observation of FW under a polytope constraint where FW often has a zig-zag trajectory.


typos:
In Table 1, the "computation" in the 'second' column.

**Time Spent Reviewing:**

2

---

> ### Author Response · Authors · 2021-08-08
> **Response to Reviewer DsMv**
>
> Thanks for the valuable comments, we find them useful to improve the quality of this work. Thanks also for pointing out the typos. We will correct them in the revised version.
>
>
> Q: How is NGD related with HFW on the l2 norm ball?
>
> A: Indeed, on l2 norm constraints, HFW is similar to normalized gradient descent (NGD). We will add this discussion in the revised paper. However, there is a subtle difference between HFW and NGD. Consider $||x || \leq 1$ for example. The update of NGD is $x_{k+1} = Proj( x_k - \eta_k \frac{g_k}{\|g_k \|} )$. This implies that when projection is in effect, $x_{k+1}$ will lie on the boundary of the l2 norm ball. While for FW, the update is $x_{k+1} = (1-\eta_k) x_k + \eta_k \frac{g_k}{\|g_k \|}$. Due to the convex combination in the HFW update, it is unlikely to have $x_{k+1}$ on the boundary, though it can come arbitrarily close.

---

### Official Review · Reviewer_BKNr · 2021-07-16

**Rating:** 8
**Confidence:** 3

**Summary:**

The authors provide an analysis of Frank Wolfe (FW) algorithms with heavy ball momentum. After a recap of standard FW including common assumptions, choices of step size and convergence guarantees, they introduce FW with heavy ball momentum. Different variants for step size and weighting parameter are considered and it is shown that using heavy ball momentum stabilizes convergence in terms of an improved rate. Subsequently, the authors consider FW with heavy ball momentum prove that this leads to further improvement of the convergence rate. Finally, numerical results for binary classification on multiple datasets and matrix completion are reported.

**Limitations And Societal Impact:**

Yes.

**Main Review:**

From my point of view, this is a very good paper. The presentation is clear and concise and, to the best of my knowledge, the reported results and proof techniques are novel and correct. FW algorithms are relevant for various applications and incorporating heavy ball momentum seems natural. This paper fills a gap in the sense that improved convergence guarantees for this case are shown. The reported numerical results support the theoretical claims. I have no further concerns.

**Time Spent Reviewing:**

3

---

> ### Author Response · Authors · 2021-08-08
> **Response to Reviewer BKNr**
>
> Thank you for reviewing this submission, and also for the favorable recommendation.
>
> Q: How about social impact?
>
> A: As the topic is centered around analytical perspective of FW, we believe it has no direct social impact.

---

### Official Review · Reviewer_Tvht · 2021-07-20

**Rating:** 8
**Confidence:** 4

**Summary:**

This work presents the heavy ball frank wolfe (HFW) method and shows that it establishes a $\mathcal{O}(1/K)$ last-iterate rate on a primal-dual gap (and hence also on the primal function-value suboptimality). In contrast, prior work established a $\mathcal{O}(1/K)$ last-iterates rate on the primal suboptimality or a $\mathcal{O}(1/K)$ best-iterate rate on the primal-dual gap. The work also presents variations of the HFW method and brief experiments.

**Main Review:**

In my view, the main and most original contribution of this work is the proof technique used in Theorem 1. The conciseness and simplicity of the proof of Theorem 1 make the technique not only elegant but likely useful in other related Frank-Wolfe type setups. The remaining content of the paper, in my view, is ancillary.

More specifically, I feel that the core proof idea of defining the dual error via a weighted average of supporting hyperplanes and having this weighted average of gradients correspond to momentum in the FW algorithm is a very nice and original idea. In hindsight, this is a simple and perhaps natural idea, since the averaging of gradients is reminiscent of dual averaging, but good ideas are often obvious is hindsight.

Granted, I have not written a Frank-Wolfe paper myself so I do not claim full expertise in this area. However, I have read many Frank-Wolfe-type papers, and this technique feels original and worthy publication to me.

**Time Spent Reviewing:**

4

---

> ### Author Response · Authors · 2021-08-08
> **Response to Reviewer Tvht**
>
> We appreciate the time and effort spent for reviewing this submission, and also for recognizing its merits. Indeed, the core of this work is Theorem 1, based on which other results are developed. Thanks also for pointing out the relation of HFW with dual averaging. Due to space limits, we in fact discuss this point in Appendix B.8

---

### Decision · Program_Chairs · 2021-09-27

**Decision:**

Accept (Poster)

**Comment:**

While the paper received a relatively high score by the reviewers and it does contain some interesting ideas, I am not not quite convinced that the merit justifies acceptance at NeurIPS. Not wanting to overrule the reviewers I recommend weak acceptance.